# EffComp: Efficient Prompt Compression via Hybrid Reinforcement & Supervised Learning

## Abstract

Prompt compression is aimed at reducing input prompt lengths to enable cheaper and faster LLM predictions. However, existing prompt compression methods are often limited by modest compression gains, a risk of hallucination, and/or high compression latency. This paper proposes EffComp, an **Eff**icient prompt **Comp**ression framework using a hybrid reinforcement and supervised learning approach for RAG-based open-domain question answering (QA). EffComp employs BERT-style document reranker and sentence selector models to allow fast extractive prompt compression at the sentence level. Its extractive nature prevents hallucinations in the compressed prompts. Additionally, the training process is designed to optimize the compression ratio while preserving LLM accuracy. Experiments on four open-domain QA datasets demonstrate that EffComp outperforms state-of-the-art prompt compression methods in terms of prediction accuracy and achieves competitive compression ratios (up to 78.4x) with minimal latency, making it practical for real-world applications.

## 1 Introduction

Retrieval-augmented generation (RAG) enhances large language models (LLMs) by grounding their predictions in retrieved external documents to reduce hallucinations and address domain gaps (Lewis et al., 2020; Guu et al., 2020; Izacard & Grave, 2021). However, the retrieved documents could often be long, redundant, and partially irrelevant (Barnett et al., 2024; Lertvittayakumjorn et al., 2025). The resulting lengthy augmented prompts not only undermine the quality of LLM predictions (Shi et al., 2023) but also increase computation costs and inference latency (Duman Keles et al., 2023). Therefore, recent research has proposed prompt compression methods to reduce the input prompt lengths (e.g., by strategically compressing the retrieved contexts) to address these efficiency and quality problems (Li et al., 2025b; Wingate et al., 2022; Chevalier et al., 2023).

Despite these advancements, existing methods face several challenges and trade-offs. Abstractive prompt compressors generate a shorter context from a given long one, but they risk fabricating details or answering the questions themselves (Xu et al., 2024; Alansari & Luqman, 2025). Extractive prompt compressors, by contrast, preserve grounding by selecting and keeping the most important parts of the original context, but they often yield only modest compression ratios, e.g., 2x–7x in Jiang et al. (2024); Pan et al. (2024); Fei et al. (2025); Zhao et al. (2025a;b). Furthermore, some compressors rely on large models or iterative algorithms, which can cause substantial latency and offset the benefits of compression (Yoon et al., 2024).

To overcome these challenges, we propose EffComp, a novel prompt compression framework for open-domain question answering (QA). The framework aims to boost the efficiency of prompt compression and subsequent LLM inference while maintaining (or even improving) the quality of the final output. EffComp consists of two main steps. The first step reranks and filters the retrieved documents using a reranker. Then the second step selects important sentences in the remaining documents and concatenates them to form a compressed context for LLM inference. The novelty of EffComp also lies in the training process of its sentence selector. This process first pre-trains the model on a QA dataset and then fine-tunes it using a hybrid reinforcement and supervised learning approach that leverages signals from the LLM. We compare our framework against reranker-based and other state-of-the-art prompt compression methods on four QA datasets. The experiments show

that EFFCOMP achieves competitive or higher QA accuracy and a strong compression ratio, outperforming other compression methods. Overall, our contribution is threefold:

- We introduce EFFCOMP, a sentence-level extractive prompt compression framework for RAG-based open-domain QA. Our framework's efficiency stems from its use of compact document reranker and sentence selector models. Furthermore, the sentence selector is trained using a hybrid reinforcement and supervised learning approach to optimize the trade-off between compression ratio and LLM predictive performance. (See Section 3.)
- We evaluate EFFCOMP on four QA datasets (Natural Questions, TriviaQA, HotpotQA, and 2WikiMultiHopQA) with two target models, demonstrating that it achieves substantially high extractive compression while maintaining strong QA performance. (See Section 4.)
- We conduct additional analyses to understand the latency, generalizability, factual consistency between extractive and abstractive compression methods, the retention improvements from Phase II GRPO, and potential points of failure of EFFCOMP (see Section 5).

## 2 PROBLEM FORMULATION

In a RAG-based open-domain QA setting, let $q$ denote a question (so-called an input query) and $D = \langle d_1, d_2, \ldots, d_n \rangle$ denote a list of $n$ retrieved documents ordered by the retrieval scores decreasingly. The full context before compression $\mathbb{D}$ is the concatenation of all $d_i \in D$ and has the length of $|\mathbb{D}|$ tokens. RAG combines $q$ and $\mathbb{D}$ to be an input prompt and feeds it to a target LLM $M$, which then outputs $M(q, \mathbb{D})$ to answer the question $q$. Prompt compression aims to find a compressed context $\mathbb{C}$ such that $|\mathbb{C}| < |\mathbb{D}|$ and $M(q, \mathbb{C})$ is a correct answer of $q$. In this paper, we focus on two correctness metrics. First, the accuracy (Acc) for a given context $X$ equals 1 if $M(q, X)$ contains the reference answer of $q$ (denoted as $y_q$) after text normalization[1]; otherwise, 0. However, there is one exception: if the reference answer is "yes", "no", or "noanswer", we follow Yoon et al. (2024) and Jung et al. (2024) and use an exact match between $M(q, X)$ and $y_q$ instead. For the second metric, F1 of $X$ is the token-level F1 measure of $M(q, X)$ compared to the reference $y_q$. Following Jiang et al. (2024), we define the compression ratio as $|\mathbb{D}|/|\mathbb{C}| = 1/\tau$. EFFCOMP aims to optimize the correctness metrics while achieving a competitively high compression ratio. We decide not to aggressively optimize $1/\tau$ as it might encourage the lightweight compressor to identify the answer itself.

## 3 EFFCOMP: THE EFFICIENT PROMPT COMPRESSION FRAMEWORK

EFFCOMP compresses the retrieved context in two main steps, one at the document level and the other at the sentence level, as illustrated in Figure 1.

**Step 1: Document reranking.** Given $n$ retrieved documents $\langle d_1, d_2, \ldots, d_n \rangle$, an off-the-shelf document reranker is used to compute a relevancy score for each document $d_i$. The reranker takes the question $q$ and a document $d_i$ as input and outputs a score indicating the relevance of $d_i$ to $q$. Only top $k$ documents with the highest scores are kept and then sorted by their scores in decreasing order. This is performed because evidence appearing earlier in an input prompt is more likely to receive stronger attention from LLMs (Tang et al., 2025). To ensure the efficiency of this step, we experiment with three BERT-style document rerankers, including BGE (Xiao et al., 2023), Jina (Günther et al., 2023), and GTE (Zhang et al., 2024). All of them are relatively lightweight models and can compute the relevancy scores in batches.

**Step 2. Sentence selection.** EFFCOMP concatenates the reranked documents from Step 1 and splits the entire concatenated text into a list of sentences $S = \langle s_1, s_2, \ldots, s_m \rangle$. After that, a sentence selector model is applied to identify sentences that should be kept in the final compressed context (rather than discarded). Our sentence selector also relies on the BERT architecture (Devlin et al., 2019) for high efficiency. It takes the question $q$ and all the $m$ sentences in $S$ as input and predicts either 0 or 1 for each sentence $s_i$. Specifically, the input representation $I(q, S)$ at the token level is

$$[\texttt{CLS}]\, q_1 q_2 \ldots q_l\, [\texttt{SEP}]\, [\texttt{SEN}]\, t_{11} t_{12} \ldots t_{1h_1}\, [\texttt{SEN}]\, t_{21} \ldots t_{2h_2} \ldots [\texttt{SEN}]\, t_{m1} \ldots t_{mh_m}\, [\texttt{SEP}]$$

---

[1]Following the evaluation script for SQuAD version 2.0 (Rajpurkar et al., 2018), we perform text normalization by lowercasing the text, removing punctuation and articles, and collapsing extra whitespace.

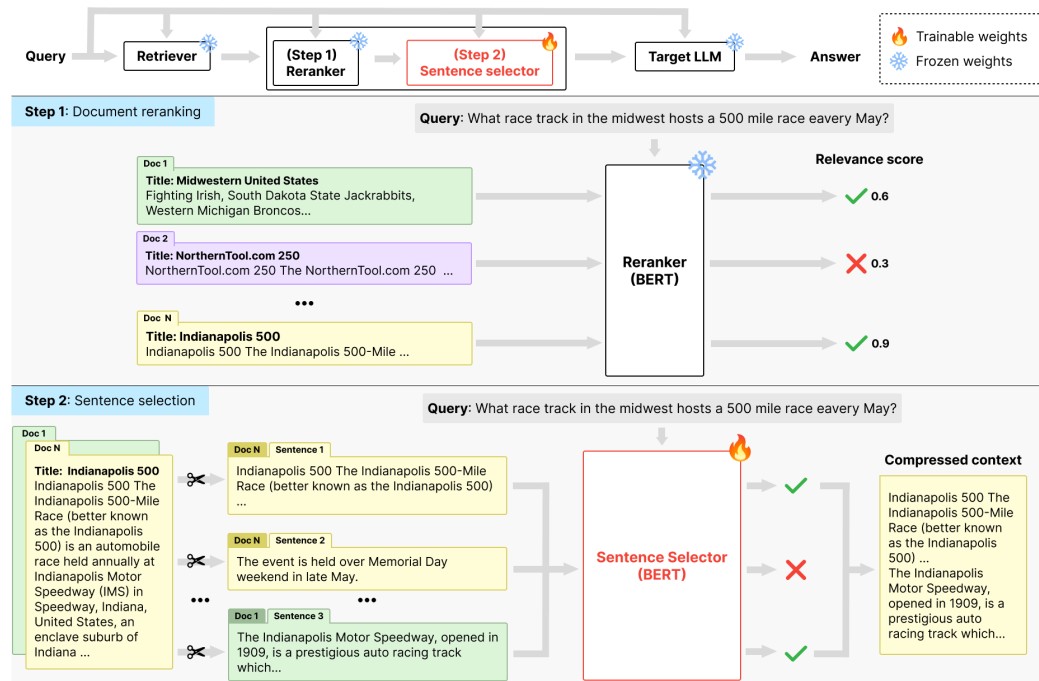

Figure 1: The inference workflow of EFFCOMP. The box with Step 1 and Step 2 at the top illustrates where EFFCOMP is in the prediction pipeline. In Step 1, retrieved documents are filtered, reordered, and concatenated. In Step 2, they are decomposed into sentences, from which necessary ones are selected and concatenated to form the compressed context for LLM prediction.

where $t_{ij}$ is the $j$-th token of $s_i$ (which has $h_i$ tokens in total) and $q_j$ is the $j$-th token of $q$ (which has $l$ tokens in total). Apart from the standard [CLS] and [SEP] tokens, we prepend a special [SEN] token to each sentence to support sentence-level classification. The sentence selector model outputs predictions at every token position, but only the outputs at the [SEN] tokens are used, indicating whether each corresponding sentence is retained or discarded. The retained sentences are concatenated without changing their order to be the compressed context $\mathbb{C}$. See Appendix B for more details.

Overall, the design of EFFCOMP offers several advantages. First, the finest level of selection is at the sentence level (rather than the token level). This renders a compressed context that remains understandable even under extreme compression and aligns naturally with how target LLMs were trained. Furthermore, the sentence selector in Step 2 considers all remaining context sentences simultaneously, enabling it to discard useful but redundant sentences to maximize the compression ratio. This simultaneous processing remains efficient thanks to Step 1, which already discards a significant amount of irrelevant information at the document level.

The remainder of this section explains the training process of the BERT-based sentence selector, which has two phases. While Phase I (pretraining) prepares the model to recognize sentences that are likely relevant to the query, Phase II (finetuning) further adapts the model to the target LLM.

## 3.1 PHASE I: PRETRAINING

**Training set construction.** We adapt training examples from an open-domain QA dataset to be our pretraining examples. To explain, each pretraining example consists of a question $q$ from the dataset and a list of retrieved documents $D$ with $|D| = 30$. We concatenate all the documents in $D$, split the concatenated text into sentences $S$, and format them (together with $q$) as input to the sentence selector. Inspired by document-level label assignment in CompAct (Yoon et al., 2024), we assign a pseudo-label to each sentence using the following rule: a sentence is labeled as *retained* if it contains the ground-truth answer string; otherwise, it is labeled *discarded*. In Phase I, we train

the sentence selector to predict these labels. Note that we exclude two types of examples from the pretraining phase: (1) examples with $|S| > 5120$ tokens, as these cases rarely happen because of the small $k$ in the document reranking step (2) examples with ground-truth answers such as "yes", "no", or "noanswer", as they do not correspond to concrete evidence in the retrieved text and can therefore cause misleading supervision.

**Training objective.** The training objective is defined at the positions of the [SEN] tokens. For each sentence $s_i \in S$, the action $a_i^*$ (retained or discarded) is predicted at the position of its preceding [SEN] token. A masked cross-entropy loss is applied such that only [SEN] positions contribute to the objective, while all other tokens are ignored. This encourages the model to learn sentence-level decisions by treating each [SEN] token as the supervision point for its corresponding sentence.

### 3.2 PHASE II: FINETUNING

The pretrained model from Phase I is not yet optimal because the pseudo-labels used for pretraining are based on a simple string matching heuristic. Specifically, some sentences that do not contain the ground-truth answer may still be useful or even necessary for the target LLM, especially for questions requiring multi-hop or cross-document reasoning. Conversely, a sentence containing the ground-truth answer may not be necessary if its content is irrelevant to the question or if it is redundant with another selected sentence. Therefore, Phase II addresses these issues by using a hybrid reinforcement learning (RL) and supervised learning (SL) approach to fine-tune the model. As noted in TACO-RL (Shandilya et al., 2024), RL only fine-tuning is computationally expensive and time-consuming. To reduce this burden while still benefiting from RL, we adopt a hybrid strategy that utilizes the trajectories from RL efficiently. In each epoch, we update the model weights using an RL loss followed by an SL loss, as explained next. We further analyze the training-time implications of this hybrid approach in Appendix D.9.

#### 3.2.1 REINFORCEMENT LEARNING

We consider our sentence selector model as a policy model $\pi_\theta$ and use reward signals derived from the target LLM and ground-truth QA answers to optimize it.

**Problem formulation.** We formulate our task as a contextual multi-armed bandit problem (Jung & Kim, 2024; Lu et al., 2010), which terminates after a single decision step. A state $\sigma$ is the input to the policy model $I(q, S)$ where $q$ is a question and $S$ is the list of context sentences from the top-$k$ reranked documents. The policy $\pi_\theta$ takes $\sigma$ as input and produces output probability distributions for all input tokens, but, consistent with the pretraining phase, only the probability distributions at the [SEN] tokens are used to derive binary actions $a_i \in \{0, 1\}$ indicating whether the corresponding sentences are retained (1) or discarded (0). With $m$ sentences in $S$, one state produces multiple actions $A = \langle a_1, a_2, \ldots, a_m \rangle$ in parallel, where each action is sampled from the categorical distribution

$$a_i \sim \text{Categorical}\big(\pi_\theta(\cdot \mid \sigma, i)\big), \quad \forall i \in \{1, \ldots, m\}. \tag{1}$$

The compressed context $\mathbb{C}$ is then formed by concatenating all retained sentences. The target LLM $M$ produces $M(q, \mathbb{C})$, and a reward $r$ for $A$ is then computed. A trajectory, therefore, consists of a single state $\sigma$, the full list of binary actions $A$, and the resulting reward $r$.

**Reward function.** We define the reward $r$ as follows.

$$r = \begin{cases} \alpha + (1 - \alpha) \cdot (1 - \tau), & \text{if } \text{Acc}(M(q, \mathbb{C}), y_q) = 1 \\ -(1 - \alpha) \cdot (1 - \tau), & \text{otherwise} \end{cases} \tag{2}$$

where $\tau = |\mathbb{C}|/|\mathbb{D}|$ and $y_q$ is the ground-truth answer of $q$. This reward jointly maximizes QA accuracy and compression, with their relative importance controlled by the weighting factor $\alpha \in (0, 1)$. During validation, we primarily use this reward to select the best model checkpoint for evaluation. For a detailed analysis of how different values of $\alpha$ influence this checkpoint selection process, see Appendix D.11. In practice, we prefer a relatively high $\alpha$ to avoid extreme compression at the expense of accuracy.

**Optimization.** To train the policy $\pi_\theta$, we draw inspiration from Group Relative Policy Optimization (GRPO) (Shao et al., 2024), a policy gradient method that removes the value function used in

PPO (Schulman et al., 2017) and instead efficiently estimates advantages by normalizing rewards within a group of sampled trajectories.

Given a group size $b$, for each state, we sample a group of actions $\mathbf{A} = \{A_1, \ldots, A_b\}$ from $\pi_\theta$, each producing a compressed context $\mathbb{C}_i$ and reward $r_i$. The reward set $R = \{r_1, \ldots, r_b\}$ is z-normalized in group to yield relative advantages $Z = \{z_1, \ldots, z_b\}$ where $z_i = \frac{1}{\text{std}(R)}(r_i - \text{mean}(R))$, used for updating the policy.

To measure the degree of policy change, we define $\pi_\theta(A \mid \sigma)$ as the geometric mean of the probabilities across all sampled binary actions in $A$:

$$\pi_\theta(A \mid \sigma) = \left( \prod_{i=1}^{m} \pi_\theta(a_i \mid \sigma) \right)^{\frac{1}{m}} \tag{3}$$

Our reinforcement learning objective is then defined as

$$\mathcal{L}_{\text{RL}}(\theta) = -\mathbb{E}\left[ \min\left( \frac{\pi_\theta(A \mid \sigma)}{\pi_{\theta_{\text{old}}}(A \mid \sigma)} z, \ \text{clip}\left( \frac{\pi_\theta(A \mid \sigma)}{\pi_{\theta_{\text{old}}}(A \mid \sigma)}, \ 1 - \epsilon, \ 1 + \epsilon \right) z \right) \right], \tag{4}$$

where $\pi_\theta$ and $\pi_{\theta_{\text{old}}}$ denote the current and old policy models, respectively, and $\epsilon$ is a clipping hyperparameter, typically set to $0.2$.

Additionally, we compute the Shannon entropy (Shannon, 1948) of the policy $\pi_\theta$ at the state $\sigma$ (considered only at the [SEN] positions) as

$$H(\pi_\theta(\cdot \mid \sigma)) = \mathbb{E}\left[ -\frac{1}{m} \sum_{i=1}^{m} \sum_{a_i \in \{0,1\}} \pi_\theta(a_i \mid \sigma) \log \pi_\theta(a_i \mid \sigma) \right] \tag{5}$$

The final training loss is

$$\mathcal{L}_{\text{total}}(\theta) = \mathcal{L}_{\text{RL}}(\theta) - \lambda H(\pi_\theta(\cdot \mid \sigma)), \tag{6}$$

where $\lambda$ controls the trade-off between exploitation and exploration. In practice, we set $\lambda \geq 0.1$, which encourages more exploration.

**Reward normalization scenarios.** It is worth noting that the reward normalization (z-norm) by GRPO naturally guides our compression. As illustrated in Figure 2, when at least one $A_i$ results in a correct LLM output, the policy will learn to increase compression. In contrast, when all $A_i \in \mathbf{A}$ lead to incorrect LLM outputs, it will reduce compression to preserve more information. This reward normalization effect both prevents extreme behaviors and enables the framework to learn a stable trade-off between accuracy and compression.

**Training set construction.** For each QA training example $(q, y_q)$, we perform Step 1 (document reranking) to obtain context sentences $S$ from the top-$k$ documents to form the input $I(q, S)$. However, we exclude some training examples. First, we exclude examples that the target LLM cannot answer correctly even with all sentences in $S$ provided, as further compression is unlikely to help. Given our reward function, including such examples would bias the policy to always retain all sentences, making the training inefficient. Second, we exclude examples that the target LLM can answer correctly even without any context. Keeping these examples would bias the policy toward discarding all sentences, resulting in non-grounded LLM outputs. This example filtering step ensures that the training set only contains examples where context is necessary and compression choices truly matter. Hence, appropriate decompression can recover missing evidence, while excessive compression inevitably removes the information required to answer.

### 3.2.2 Supervised Learning

During GRPO, we sample multiple $A_i$ for each state $\sigma$. To enhance training efficiency, we introduce a memory $C$ to store the best trajectory of each state observed so far. (In other words, it stores the best list of binary actions discovered so far for each example.) For each state $\sigma$, we store a trajectory $(\sigma, A_i, r_i)$ in $C$ when $r_i > 0$ and update it when we find another trajectory $(\sigma, A_j, r_j)$ where $r_j \geq r_i$. After the RL update of each epoch, these best trajectories in $C$ are used as silver labels to finetune the policy (i.e., the sentence selector) in a supervised learning manner. We apply the cross-entropy

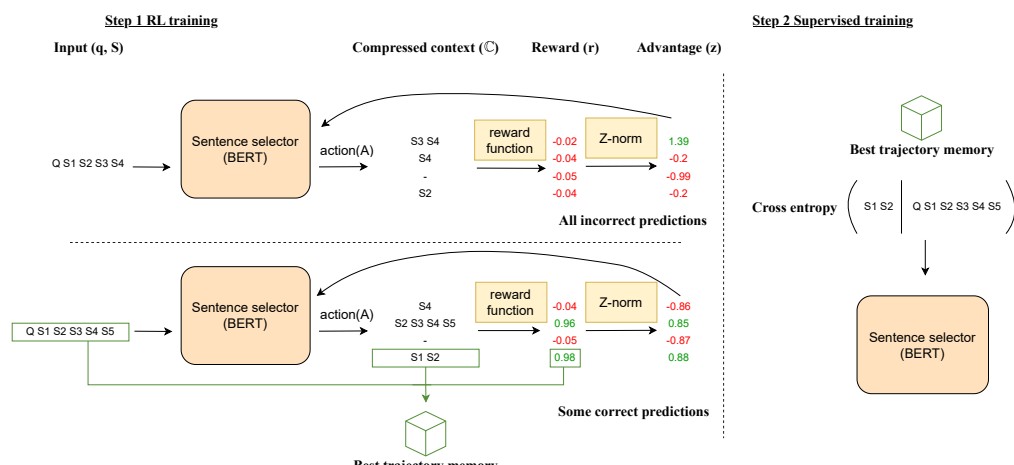

Figure 2: Overall finetuning pipeline for the sentence selector within a single epoch. Reinforcement learning is applied first, followed by supervised learning. Reward normalization encourages higher compression when some predictions are correct and lower compression when all predictions fail.

loss at the [SEN] positions using the stored actions as labels. To avoid overfitting, each example in $C$ is used to train the policy only once until a new trajectory of the example is updated in the memory. The detailed algorithm of our approach can be found in Appendix C. We also include an ablation study comparing SL only, RL only, and our hybrid training setup in Appendix D.8.

# 4 EXPERIMENTS

We trained and evaluated EFFCOMP on four open-domain QA datasets: HotpotQA (HQA; Yang et al. (2018)), 2WikiMultiHopQA (2WikiQA; Ho et al. (2020)), Natural Questions (NQ; Kwiatkowski et al. (2019)), and TriviaQA (TQA; Joshi et al. (2017)), using the training sets for pretraining and finetuning and the development sets for evaluation.

## 4.1 EXPERIMENTAL SETUP

**Implementation details.** Following prior work (Yoon et al., 2024; Xu et al., 2024), we implemented the retrieval system using Contriever (Izacard et al., 2021), fine-tuned on MSMARCO (Bajaj et al., 2018). For each question, we retrieved $n = 30$ documents from the 2018 Wikipedia corpus (Karpukhin et al., 2020). Additionally, we used Gemma-2-9B-IT[2] (Team, 2024) as the target LLM. For EFFCOMP, we tried three document rerankers, including BGE[3] (Xiao et al., 2023), Jina[4] (Günther et al., 2023), and GTE[5] (Zhang et al., 2024), to rerank and select top-$k$ documents ($k = 10$) in Step 1. For Step 2, we used ModernBERT-base[6] (Warner et al., 2024) as the sentence selector and used the BGE reranker to prepare input documents for finetuning the sentence selector. We also evaluated EFFCOMP without Step 1, which passed sentences from $n$ documents to Step 2 without reranking. This variant is denoted as "No reranker" in the result tables.

**Baselines.** We compared EFFCOMP against several types of baselines. The simplest one was "Raw Documents", which inputs the $n$ retrieved documents to the target LLM without compression. Next, reranker-based filtering baselines used a reranker (BGE, Jina, or GTE) to select top-$k$ documents to feed into the target LLM. This is equivalent to EFFCOMP without Step 2. Furthermore, we compared EFFCOMP with state-of-the-art compression frameworks, including two extractive methods

---

[2]https://huggingface.co/google/Gemma-2-9B-IT
[3]https://huggingface.co/BAAI/bge-reranker-base
[4]https://huggingface.co/jinaai/jina-reranker-v2-base-multilingual
[5]https://huggingface.co/Alibaba-NLP/gte-reranker-modernbert-base
[6]https://huggingface.co/answerdotai/ModernBERT-base

| Method | HQA | | | 2WikiQA | | | NQ | | | TQA | | |
|---|---|---|---|---|---|---|---|---|---|---|---|---|
| | $1/\tau$ | Acc | F1 | $1/\tau$ | Acc | F1 | $1/\tau$ | Acc | F1 | $1/\tau$ | Acc | F1 |
| Raw Documents | 1x | 34.6 | 42.3 | 1x | 29.1 | 32.9 | 1x | 30.1 | **38.0** | 1x | 77.4 | 77.2 |
| *Reranker-based methods* | | | | | | | | | | | | |
| BGE reranker | 3x | 35.2 | 42.1 | 3x | 23.8 | 27.6 | 3x | 29.6 | 35.4 | 3x | 79.9 | 77.1 |
| Jina reranker | 3x | 34.6 | 41.5 | 3x | 24.4 | 28.2 | 3x | 30.6 | 36.7 | 3x | 80.1 | 77.1 |
| GTE reranker | 3x | 34.7 | 41.6 | 3x | 23.9 | 27.5 | 3x | 30.6 | 36.4 | 3x | 80.4 | **77.4** |
| *Compression frameworks* | | | | | | | | | | | | |
| LongLLMLingua (5x) | 5.1x | 31.8 | 38.2 | 5.1x | 21.8 | 25.8 | 5.1x | 25.7 | 31.7 | 5.1x | 78.4 | 75.1 |
| LongLLMLingua (19x) | 19.1x | 28.7 | 35.7 | 19.3x | 20.5 | 24.5 | 18.9x | 20.7 | 26.8 | 19.1x | 73.9 | 71.7 |
| Recomp-Extractive | 29.2x | 25.6 | 31.9 | 29.3x | 17.4 | 21.4 | 29.8x | 22.8 | 28.2 | 32x | 73.1 | 70.4 |
| Recomp-Abstractive | **132.1x** | 27.9 | 34.0 | **126.9x** | 21.0 | 25.1 | **48.3x** | 27.6 | 33.1 | **139x** | 73.0 | 70.7 |
| CompAct | 48.7x | 31.2 | 36.8 | 54.7x | 16.8 | 20.1 | 44.2x | 27.8 | 32.8 | 50.2x | 77.6 | 74.3 |
| *Our work:* EFFCOMP | | | | | | | | | | | | |
| No reranker | 26.8x | **37.4** | **44.3** | 16.1x | **29.5** | **33.4** | 20.1x | **30.9** | 37.2 | 48.4x | 78.9 | 76.8 |
| BGE reranker | 39.5x | 36.3 | 43.0 | 30.3x | 25.4 | 29.1 | 29.1x | 29.8 | 35.4 | 59.5x | 80.1 | 76.8 |
| Jina reranker | 38.5x | 35.8 | 42.5 | 29.4x | 25.5 | 29.4 | 34x | 29.9 | 35.7 | 78.4x | 80.1 | 76.7 |
| GTE reranker | 40.4x | 35.5 | 42.4 | 29.4x | 25.4 | 29.1 | 33.2x | 30.5 | 36.0 | 64.3x | **80.5** | 77.1 |

Table 1: Results on four open-domain QA benchmarks. We report the mean of compression ratio $(1/\tau)$, accuracy (Acc), and token-level F1 score for **Gemma-2-9B-IT** as the target model. For all of these metrics, higher values are better. The best results in each column are marked in bold.

(Recomp-Extractive (Xu et al., 2024) and LongLLMLingua (Jiang et al., 2024)) and two abstractive methods (Recomp-Abstractive (Xu et al., 2024) and CompAct (Yoon et al., 2024)). Further implementation details of the baselines are provided in Appendix B.6.

## 4.2 RESULTS

Table 1 reports the results on three metrics: compression ratio $(1/\tau)$, accuracy (Acc), and F1 score. Compared to using the Raw Documents, EFFCOMP with rerankers achieved competitive or sometimes higher Acc and F1 with the compression ratios of 29.1x–78.4x. For example, on HQA, EFFCOMP with BGE reranker achieved 36.3% accuracy with the compression ratio of 39.5x, while the Raw Documents got 34.6% accuracy with no compression (1x). This demonstrates that context compression, if done appropriately, can enhance rather than degrade the quality of LLM predictions. Without the rerankers, EFFCOMP got significantly lower compression ratios but usually slightly higher Acc and F1 (e.g., 26.8x compression and 37.4% Acc on HQA). Compared to reranker-based methods, EFFCOMP with the same reranker achieved significantly higher compression ratios (e.g., from 3x to 59.5x–78.4x on TQA) and competitive, if not higher, Acc and F1 scores. This highlights the immense impact of Step 2 of EFFCOMP on compression ratios and indicates that our sentence selector, trained on BGE reranked data, generalized effectively to other rerankers such as Jina and GTE. Furthermore, EFFCOMP outperformed all other compression frameworks in terms of accuracy and F1. Although the compression ratios of EFFCOMP were lower than those of existing abstractive methods (i.e., Recomp-Abstractive and CompAct), EFFCOMP remained competitive by delivering superior QA quality. We show a successful case of EFFCOMP in Table 2. In this case, Recomp-Abstractive and CompAct generated compressed contexts that are factually incorrect or irrelevant to the question, leading to wrong LLM predictions. By contrast, LongLLMLingua and EFFCOMP remained correctly grounded in the retrieved contexts, but EFFCOMP used 8.6 times fewer tokens than LongLLMLingua (88 vs. 758 tokens). In conclusion, EFFCOMP achieved strong results across all QA benchmarks by keeping the LLM predictions well-grounded, consistently balancing high QA performance with substantial compression.

## 5 ADDITIONAL ANALYSES

We further analyze EFFCOMP through five research questions.

**RQ1: How well does EFFCOMP generalize to a different dataset or target LLM?** To test dataset generalization, we trained EFFCOMP on HotpotQA and evaluated it on the other QA datasets. As

| Question (from HotpotQA): Stein had a part in the 1994 superhero fantasy film directed by whom |
| --- |
| **Ground-truth answer**: Charles Russell |

| Method | $1/\tau \uparrow$ | $|\mathbb{C}| \downarrow$ | Compressed Context | LLM Prediction |
| --- | --- | --- | --- | --- |
| Long-LLMLingua | 5.01x | 758 | The Mask (film) The Mask is a 1994 American fantasy superhero comedy film directed by Charles Russell, produced by Bob Engelman [...] Ben Stein Benjamin Jeremy (born November 25, 1944) is an American writer, [...] He is well- screen econom teacher "Ferris Bueller's Day Off" (1986) and as Dr. Arthur Neuman in "The Mask" (1994)Son [...] | Charles Russell ✓ |
| Recomp-Abstractive | 189.75x | 20 | Ben Stein had a part in the 1994 superhero fantasy film "The Shadow" directed by Russell Mulcahy. | Russell Mulcahy ✗ |
| CompAct | 49.93x | 76 | The Shadow is a 1994 American superhero film [...] directed by Russell Mulcahy, that stars Alec Baldwin. The film co-stars John Lone, Penelope Ann Miller, Ian McKellen, Jonathan Winters, Peter Boyle, and Tim Curry. It was released to theaters on July 1, 1994 and received generally mixed reviews. | Russell Mulcahy ✗ |
| EFFCOMP (BGE) | 43.13x | 88 | He is most well-known on screen as the economics teacher in "Ferris Bueller's Day Off" (1986) and as Dr. Arthur Neuman in "The Mask" (1994) and "Son of The Mask (film) The Mask is a 1994 American fantasy superhero comedy film directed by Charles Russell, produced by Bob Engelman [...] | Charles Russell ✓ |

Table 2: Examples of compressed contexts and predictions from LongLLMLingua, Recomp-Abstrctive, CompAct, and EFFCOMP (with BGE reranker). $|\mathbb{C}|$ denotes the number of tokens of the compressed context. ✓ indicates a correct prediction, while ✗ indicates an incorrect prediction.

| Method | 2WikiQA | | | NQ | | | TQA | | |
| --- | --- | --- | --- | --- | --- | --- | --- | --- | --- |
| | $1/\tau$ | Acc | F1 | $1/\tau$ | Acc | F1 | $1/\tau$ | Acc | F1 |
| EFFCOMP (BGE) (in-distribution) | 30.3x | 25.4 | 29.1 | 29.1x | 29.8 | 35.4 | 59.5x | 80.1 | 76.8 |
| EFFCOMP (BGE) (out-of-distribution) | 19.4x | 25.0 | 28.9 | 14.1x | 29.0 | 34.8 | 49.6x | 79.5 | 76.2 |

Table 3: Cross-dataset generalization of EFFCOMP with Gemma-2-9B-IT as the target LLM. The first row reports the results when EFFCOMP was trained on each target dataset (in-distribution testing). The second row reports the results when EFFCOMP was trained on HotpotQA and applied to the target datasets (out-of-distribution testing).

shown in Table 3, although in-distribution testing achieved better QA performance, the gaps between in-distribution and out-of-distribution results were less than 1% absolute for both Acc and F1, indicating that EFFCOMP generalizes relatively well in terms of QA performance. However, compression ratios noticeably decreased under out-of-distribution evaluation. This reflects the reward design of EFFCOMP, which encouraged context decompression when the sentence selector was unsure.

Additionally, to test generalization to another target LLM, we fed the compressed contexts from the methods in Table 1 to Llama-3.1-8B-Instruct (Grattafiori et al., 2024) instead of Gemma-2-9B-IT, which was used to finetune EFFCOMP. The results show that EFFCOMP attained the highest accuracy across all baselines in four datasets and the highest F1 in three out of the four datasets. The generalizability of EFFCOMP is partly due to its operation at no finer than the sentence level, ensuring that the compressed contexts remain readable and not too specific to the target LLM used for finetuning. Full results regarding generalizability can be found in Appendices D.1 and D.2.

**RQ2: How does EFFCOMP affect inference latency?** Table 4 reports the compression latency, the target LLM reading latency, and the total latency, all averaged from 500 HotpotQA examples with a batch size of 1. Compared to no compression, EFFCOMP introduced only a small overhead during compression while substantially reducing the LLM reading time (up to 55.4% decrease latency), resulting in the lower overall latency (up to 53.3% decrease latency with Jina reranker). EFFCOMP is among the fastest methods, together with Recomp-Extractive, demonstrating their potential for use in user-facing applications.

**RQ3: Does our extractive compression method (EFFCOMP) introduce fewer factual inconsistencies than abstractive compression methods?** To answer this research question, we quantitatively evaluate hallucination using a natural language inference (NLI) setup on the HotpotQA test set. For each example, the original raw context is treated as the premise and the compressed context as the hypothesis. We then apply the tasksource/ModernBERT-large-nli (0.4B) model (Sileo, 2024) to obtain NLI predictions. We focus specifically on the contradiction label, signaling that the com-

| Method | Compression (s) | LLM Read (s) | Total (s) | Avg Prompt Length |
|---|---|---|---|---|
| Raw documents | – | 1.84 | 1.84 | 3,706.81 |
| LongLLMLingua | 2.31 | 0.90 | 3.21 | 723.69 |
| Recomp-Extractive | 0.19 | 0.74 | 0.93 | 141.71 |
| Recomp-Abstractive | 2.84 | **0.71** | 3.55 | 106.62 |
| CompAct | 16.30 | 0.72 | 17.02 | **89.93** |
| EFFCOMP (BGE) | 0.08 | 0.82 | 0.90 | 487.72 |
| EFFCOMP (Jina) | **0.04** | 0.82 | **0.86** | 499.14 |
| EFFCOMP (GTE) | 0.12 | 0.82 | 0.94 | 482.72 |

Table 4: Latency per example averaged over 500 HotpotQA examples (with average compressed-prompt length), using Gemma-2-9B-IT as the target LLM. Compression is evaluated with FlashAttention 2 (Dao, 2023) enabled for CompAct, LongLLMLingua, EFFCOMP (sentence selector), the Jina reranker, and the ModernBERT reranker. Evaluations use a batch size of 1 for both compression and LLM prediction, and a batch size of 30 for the rerankers. Latencies are measured by running the full compression pipeline including tokenization, the model forward pass, and detokenization on 500 examples. Bold numbers denote the lowest metric in each column.

| Method | Contradiction cases | % Contradiction cases |
|---|---|---|
| Recomp-Abstractive | 1,728 | 23.3 % |
| CompAct | 225 | 3.0 % |
| EFFCOMP (BGE Reranker) | **42** | 0.5 % |

Table 5: NLI-detected contradiction cases on HotpotQA (7,405 examples). Lower values indicate fewer factual inconsistencies introduced during compression.

pressed text contains information inconsistent with the original context. This evaluation provides a direct quantitative measure of how often each compression method introduces factual errors. As shown in Table 5, Recomp-Abstractive yields 1,728 contradiction cases and CompAct yields 225, while EFFCOMP (BGE Reranker) produces only 42. The substantially lower contradiction count for EFFCOMP indicates that its extractive design introduces fewer factual inconsistencies than the abstractive compressors.

**RQ4: Can Phase II GRPO effectively correct the biased sentence-selection behavior introduced by Phase I pretraining?** Phase I pretraining labels only sentences containing the final answer string, which biases the policy toward selecting answer-bearing sentences while suppressing intermediate reasoning sentences. This bias is problematic for multi-hop QA tasks such as HotpotQA, where supporting facts are essential. To assess whether Phase II can overcome this issue, we compare two models: (1) a pretrained-only model trained solely with the Phase I objective, and (2) a pretrained + finetuned model trained using both Phase I and Phase II. We evaluate how well each model preserves the human annotated supporting facts in the HotpotQA test set. A supporting fact is considered preserved if at least 50% string overlap is detected between the supporting sentence and target text. The retention rate metric is defined as $\frac{\text{\# supporting sentences preserved in compressed text}}{\text{\# supporting sentences in original document}}$.

Using this metric, the pretrained-only model achieves a retention rate of 75.77%, while the pretrained + finetuned model achieves 91.09%. These results show that Phase II effectively addresses the initial bias from Phase I and substantially improves the retention of intermediate-reasoning sentences in multi-hop QA.

**RQ5: When EFFCOMP caused QA performance regression, what were the points of failure?** Though EFFCOMP (BGE) improved the accuracy of HQA from 34.6% (Raw Documents) to 36.3%, we observed that 19.6% of the examples answered correctly with the full contexts were answered incorrectly with the compressed ones. To better understand the regression, we sampled 50 of such examples and identified the points of failure in the pipeline. The distribution of failure points was as follows: the retriever (12%), the document reranker (20%), the sentence selector (10%), the target LLM (16%), and the evaluation metric (42%). Concerning the evaluation metric, accuracy naturally misses some correct answers when their format differs from the ground truth (such as "Anabolic steroids" vs. "Anabolic Androgenic Steroids", "15" vs. "fifteen", "No" vs. "No, they are not."). This indicates that semantic-based evaluation metrics like BERTScore (Zhang et al., 2020) and LLM-based auto-raters (Vu et al., 2024) should be more widely adopted in this field. The

results also imply that the document reranker and the sentence selector have room for improvement. Finetuning the reranker and pretraining the sentence selector with more informative pseudo-labels could potentially enhance their performance and are worth exploring. Interestingly, in 12% of the cases, the correct answer was not present in the full context, but the LLM still answered correctly. It then failed when using the compressed context. We hypothesize that some information in the full context, while not explicit, was still useful to the model and was inadvertently removed during compression. Research into LLM interpretability (Singh et al., 2024) may be a useful tool for investigating these cases in detail and shedding light on potential mitigation strategies.

## 6 RELATED WORK

**Prompt compression.** Li et al. (2025a) categorize prompt compression methods into two types. Soft prompt methods compress inputs into continuous vectors readable only by the target LLM (Wingate et al., 2022; Chevalier et al., 2023; Mu et al., 2023; Ge et al., 2024; Li et al., 2025c). In contrast, hard prompt methods use truncation, selection, or summarization to keep the compressed prompts human-readable (Li et al., 2023; Jiang et al., 2024; Xu et al., 2024; Yoon et al., 2024; Pan et al., 2024; Zhao et al., 2025b; Fei et al., 2025). EFFCOMP falls into the latter category as we prioritize interpretability and applicability across different target LLMs. So far, few hard prompt methods have employed reinforcement learning. Jung & Kim (2024) formulate compression as a token-level contextual bandits problem, which may lead to overly aggressive compression. Shandilya et al. (2024) mitigate this by imposing fixed-length limits with penalties. EFFCOMP, by contrast, operates at the sentence level with a tailored reward design and training set construction. This naturally regulates compression without fixed limits, allowing higher compression while preserving QA accuracy.

**Combining reinforcement learning (RL) and supervised finetuning (SFT).** Recent studies show that using SFT to train LLMs followed by RL to align them with desired behaviors effectively enhances their capabilities (DeepSeek-AI et al., 2025; Ouyang et al., 2022). Instead of using one after the other, SuperRL (Liu et al., 2025) enhances the complex reasoning ability of LLMs using RL. However, when all sampled trajectories of an example yield zero reward, leading to uninformative RL signals, they switch to SFT with high-quality offline data. In contrast, our framework addresses a different problem by focusing on improving the efficiency of the training process. Meanwhile, our framework, EFFCOMP, alleviates the zero-reward issue by using a dense reward and a relatively high $\lambda$ in Equation 6 to encourage the exploration of various action choices. It then uses the best-performing trajectories from RL (instead of offline data) for SFT before alternating between RL and SFT like this in subsequent epochs.

## 7 CONCLUSION

We propose EFFCOMP, a query-aware prompt compression framework for RAG-based open-domain QA. EFFCOMP performs context selection at the document level and then the sentence level to ensure the remaining context is minimal yet sufficient for the target LLM and remains human-readable. Our hybrid training process teaches the model to compress more when the target LLM can answer correctly and to provide more context when it is needed. Experimental results show that EFFCOMP yields high compression ratios (up to 78.4x) while maintaining or even improving the QA accuracy (up to 8.1% relative to the no-compression baseline). It can reduce total latency by about 53.3% and generalizes well with respect to QA metrics. Future work may consider improving the pretraining of the sentence selector (e.g., by using more sophisticated labels instead of binary ones or leveraging document-level relevancy annotations from the dataset, if available) and extending EFFCOMP to other tasks beyond QA.

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

## A    THE USE OF LARGE LANGUAGE MODELS

We leveraged ChatGPT-5 and Gemini 2.5 Flash to enhance grammar, refine phrasing, and optimize word choice across the paper. We also used them to help format tables and figures in LaTeX. The typical prompts we used with the LLMs were concise instructions such as "Proofread:" or "polish this text", which direct the models to refine grammar and phrasing without altering the technical content.

## B    ADDITIONAL IMPLEMENTATION DETAILS

### B.1    INPUT REPRESENTATION OF THE SENTENCE SELECTOR

The input representation of EFFCOMP's sentence selector is illustrated in Figure 3.

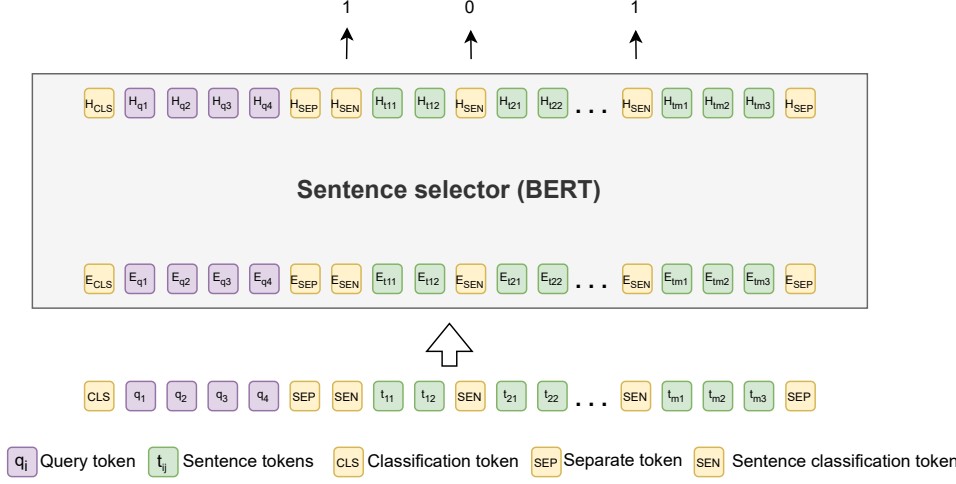

Figure 3: Input and output of the sentence selector. The input begins with the [CLS] token and the query $q$, then followed by the candidate sentences. Each sentence is prepended with a special [SEN] token. The [SEP] tokens were inserted after the query $q$ and after the last token of the last candidate sentence. The sentence selector produces token-level outputs, but only predictions at the [SEN] positions are used to determine whether each corresponding sentence is retained or discarded.

### B.2    TRAINING CONFIGURATION

Pretraining of the sentence selector was conducted using a GeForce RTX 3090 GPU, while finetuning and inference were conducted using an NVIDIA A100-SXM4-40GB GPU.

**Pretraining.**

- Learning rate: $2 \times 10^{-5}$
- Number of epochs: 2
- Weight decay: 0.01
- Batch size: 4
- Precision: bfloat16
- Optimizer: AdamW
- Loss: cross-entropy with balanced class weights

**Finetuning.**

- **Model Setup**
  - Target model: Gemma-2-9B-IT
  - Quantization: BitsAndBytes 4-bit (nf4), compute dtype: fp16
  - Sampling: disabled ($do\_sample =$ False)
  - Model maximum length: 2048
- Learning rate: $1 \times 10^{-6}$
- Number of epochs: 3
- Optimizer: AdamW
- Precision: bfloat16
- Rollout size: 128
- Group size: 8
- Backward batch size: 8
- Updates per iteration: 4
- Entropy coefficient: 0.1
- Max gradient norm: 0.5
- Clip value $\epsilon$ : 0.2
- Target KL: 0.02
- Reward weight ($\alpha$): 0.95
- Supervised loss: cross-entropy with balanced class weights

Note that we selected the best pretrained model checkpoint and the best finetuned model checkpoint based on the validation loss and the validation reward observed, respectively.

## B.3 DATA PREPROCESSING

For the pretraining data, We applied filtering by removing samples with sequence length greater than 5120 tokens and discarding samples whose answers are in the set {"yes", "no", "noanswer"}. For Natural Questions dataset, we additionally removed all instances that lack an answer. See Table 6 for dataset statistics.

Moreover, we employed the nltk sentence tokenizer to split documents into sentences.

| Dataset | Train | Validation | Test |
|---------|-------|-----------|------|
| *Base datasets* | | | |
| TriviaQA | 138,384 | – | 17,944 |
| Natural Questions | 152,148 | – | 5,499 |
| HotpotQA | 90,447 | – | 7,405 |
| 2WikiQA | 167,454 | – | 12,576 |
| *Pretraining* | | | |
| TriviaQA | 123,670 | 13,742 | – |
| Natural Questions | 133,060 | 14,785 | – |
| HotpotQA | 75,059 | 8,340 | – |
| 2WikiQA | 92,595 | 10,289 | – |
| *Finetuning* | | | |
| TriviaQA | 21,136 | 1,000 | – |
| Natural Questions | 18,687 | 1,000 | – |
| HotpotQA | 19,207 | 1,000 | – |
| 2WikiQA | 15,522 | 1,000 | – |

Table 6: Number of examples in each dataset before and after filtering. The base dataset refers to the raw data, while the pretraining and finetuning stages indicate the dataset sizes used in each phase.

### B.4    LLM Prompt Template

For both Gemma-2-9B-IT and Llama-3.1-8B-Instruct, we adopted the prompt template by Jung et al. (2024) to query the LLMs:

> You are an expert in Question Answering. Your job is to answer questions in 1 to 5 words based on the given context.
> Question: {question}
> Context: {context}
> Answer:

### B.5    Inference Configuration for Target LLMs

**Gemma-2-9B-IT.**

- Sampling: disabled (do sample = False)
- Quantization: BitsAndBytes 4-bit (nf4), dtype: float16
- Max new tokens: 10
- Model maximum length: 6144

**Llama-3.1-8B-Instruct.**

- Sampling: disabled (do sample = False)
- Quantization: BitsAndBytes 4-bit (fp4), dtype: float32
- Max new tokens: 10
- Model maximum length: 5120

### B.6    Baseline Details

This section provides additional details about the baselines in our experiments.

- **Raw Documents**: The top-30 retrieved documents are used directly without any compression.
- **Reranker-based methods**: The top-30 retrieved documents are reordered, and the top-10 documents after reordering are selected as the compressed context.
- **LongLLMLingua** (Jiang et al., 2024): A task-aware method built on Llama-2-7B[7] that ranks documents and selects tokens using contrastive perplexity scoring. We set the dynamic compression rate fixed at 0.3, with the rate set to 0.45 for the ~5× compression and 0.04 for the ~19× compression.
- **Recomp-Extractive** (Xu et al., 2024): An extractive approach that assigns scores at the sentence level using a dual-encoder model, functioning similarly to a reranker. We adopt the Contriever models released by the authors for each dataset. For 2WikiQA, where no finetuned models are available, we use the Contriever finetuned on HotpotQA. Following the previous research (Yoon et al., 2024) and similar compression ratio with our method, the top four sentences are selected as context.
- **Recomp-Abstractive** (Xu et al., 2024): An abstractive approach based on T5-large (770M) trained with summarization distillation over Natural Questions, TriviaQA, and HotpotQA. The model compresses the retrieved evidence into shorter summaries. For 2WikiQA and HotpotQA, we use the HotpotQA-trained version, while for TriviaQA and Natural Questions, dataset-specific models are applied.
- **CompAct** (Yoon et al., 2024): An iterative abstractive compression method implemented with Mistral-7B-Instruct (Jiang et al., 2023). The model repeatedly summarizes the input until the target model signals completion. The number of documents per segment is set to 5, following the original implementation.

---

[7] https://huggingface.co/NousResearch/Llama-2-7b-hf

## C DETAILED ALGORITHM

### C.1 FINETUNING THE SENTENCE SELECTOR

---

**Algorithm 1:** EFFCOMP Hybrid Reinforcement & Supervised Learning

---

**Input:** Training data $\Delta$, TargetLLM, Pretrained policy $\pi_\theta$
**Hyperparameters:** Adam optimizer (lr $= 1 \times 10^{-6}$), group size $g$=8, $\alpha$=0.95, epochs = 3, entropy coefficient $\lambda$=0.1, clip value $\epsilon$=0.2, rollout size = 128
**Output:** Optimized policy $\pi_\theta^\star$
Initialize the rollout buffer $\mathcal{B} \leftarrow \emptyset$ and the best trajectory memory $C \leftarrow \emptyset$;
**for** *example $(q, y_q, S_q) \in \Delta$* **do**
    $\sigma \leftarrow I(q, S_q)$;
    Get actions $A^\star \leftarrow \arg\max_{A \in \{0,1\}^m} \pi_\theta(\cdot \mid \sigma)$;
    Build compressed context $c \leftarrow \mathrm{Select}(S_q, A^\star)$;
    Obtain output $y \leftarrow \mathrm{TargetLLM}(q, c)$;
    Compute reward $r \leftarrow \mathrm{Reward}(y_q, y, S_q, c)$;
    UpdateMemory$(C, \sigma, \langle A^\star \rangle, \langle r \rangle)$ using Algorithm 2;
reference model $\pi_{ref} \leftarrow \pi_\theta$;
**for** *epoch = 1* **to** epochs **do**
    reference model $\pi_{ref} \leftarrow \pi_\theta$;
    **for** *example $(q, y_q, S_q) \in \Delta$* **do**
        $\sigma \leftarrow I(q, S_q)$;
        Sample actions $\mathbf{A} \leftarrow \{A_i\}_{i=1}^g \sim \pi_\theta(\cdot \mid \sigma)$ (Eq. 1);
        **for** *$i = 1$* **to** $g$ **do**
            $c_i \leftarrow \mathrm{Select}(S_q, A_i)$;
            $y_i \leftarrow \mathrm{TargetLLM}(q, c_i)$;
            $r_i \leftarrow \mathrm{Reward}(y_q, y_i, S_q, c_i)$;
        Let $R \leftarrow \{r_i\}_{i=1}^g$;
        UpdateMemory$(C, \sigma, \mathbf{A}, R)$ using Algorithm 2;
        Compute relative advantages $Z$ from $R$;
        Append $(\sigma, \mathbf{A}, Z)$ to the rollout collection $\mathcal{B}$;
        **if** *$|\mathcal{B}| \geq$ rollout size* **then**
            Update $\pi_\theta$ with final training loss (Eq. 6);
            reference model $\pi_{ref} \leftarrow \pi_\theta$;
            Reset $\mathcal{B} \leftarrow \emptyset$;
    **for** *each $(S_b, A_b) \in C$* **do**
        Update $\pi_\theta$ with supervised cross-entropy on $(S_b, A_b)$;
**return** $\pi_\theta^\star$;

---

### C.2 BEST TRAJECTORY MEMORY

---

**Algorithm 2:** EFFCOMP Update the best trajectory memory

---

**Input:** Buffer $C$ (a dictionary mapping each state $\sigma$ to its best trajectory), state $\sigma$, group actions $\mathbf{A}$, group rewards $R$
**Output:** Updated buffer $C$
**for** *$(A_i, r_i) \in (\mathbf{A}, R)$* **do**
    **if** *$r_i > 0$ and ($\sigma \notin C$ or $r_i \geq C[\sigma].\mathrm{reward}$)* **then**
        $C[\sigma] \leftarrow (A_i, r_i)$
**return** $C$

---

# D    ADDITIONAL RESULTS AND ANALYSES

## D.1    GENERALIZATION TO ANOTHER TARGET MODEL

We evaluated our method against baseline approaches under the setting where compressed contexts were provided to another target model which is Llama-3.1-8B-Instruct. Table 7 reports the results on HotpotQA, 2WikiMultiHopQA, Natural Questions, and TriviaQA.

| Method | HQA | | | 2WikiQA | | | NQ | | | TQA | | |
|---|---|---|---|---|---|---|---|---|---|---|---|---|
| | $1/\tau$ | Acc | F1 | $1/\tau$ | Acc | F1 | $1/\tau$ | Acc | F1 | $1/\tau$ | Acc | F1 |
| Raw Documents | 1x | 24.9 | 29.3 | 1x | 16.7 | 19.1 | 1x | 25.3 | 31.2 | 1x | 69.7 | 66.2 |
| *Reranker-based methods* | | | | | | | | | | | | |
| BGE | 3x | 29.6 | 33.1 | 3x | 19.2 | 21.6 | 3x | 26.6 | 31.9 | 3x | 75.3 | 69.8 |
| Jina | 3x | 29.2 | 33.3 | 3x | 19.3 | 22.0 | 3x | 28.1 | 33.2 | 3x | 75.6 | 70.2 |
| GTE | 3x | 29.0 | 32.9 | 3x | 19.2 | 21.8 | 3x | 28.4 | 33.5 | 3x | 75.9 | 70.3 |
| *Compression frameworks* | | | | | | | | | | | | |
| LongLLMLingua (5x) | 5.1x | 27.7 | 32.4 | 5.1x | 18.8 | 21.6 | 5.1x | 24.4 | 30.3 | 5.1x | 73.6 | 69.6 |
| LongLLMLingua (19x) | 19.1x | 25.6 | 30.6 | 19.3x | 18.9 | 22.0 | 18.9x | 19.7 | 25.0 | 19.1x | 69.3 | 64.6 |
| Recomp-Extractive | 29.2x | 24.0 | 27.8 | 29.3x | 16.6 | 19.4 | 29.8x | 22.3 | 26.4 | 32x | 69.7 | 63.7 |
| Recomp-Abstractive | **132.1x** | 27.8 | 32.1 | **126.9x** | 21.3 | **24.5** | **48.3x** | 26.3 | 30.4 | **139x** | 74.3 | 68.4 |
| CompAct | 48.7x | 30.5 | 33.1 | 54.7x | 18.8 | 20.9 | 44.2x | 27.9 | 31.1 | 50.2x | 75.9 | 67.9 |
| *Our work:* EFFCOMP | | | | | | | | | | | | |
| No reranker | 26.8x | 29.6 | 33.9 | 16.1x | 21.3 | 23.6 | 20.1x | 27.6 | 33.1 | 48.4x | 73.5 | 69.0 |
| BGE reranker | 39.5x | **31.3** | **35.0** | 30.3x | 21.5 | 23.8 | 29.1x | 28.5 | 33.1 | 59.5x | 76.6 | 70.3 |
| Jina reranker | 38.5x | 31.0 | 34.9 | 29.4x | **21.8** | 24.1 | 34x | 29.0 | 33.5 | 78.4x | 77.1 | **70.8** |
| GTE reranker | 40.4x | 31.0 | 34.8 | 29.4x | 21.6 | 23.9 | 33.2x | **29.4** | **34.0** | 64.3x | **77.3** | **70.8** |

Table 7: Results on four open-domain QA benchmarks. We report the mean of compression ratio ($1/\tau$), accuracy (Acc), and token-level F1 score for **Llama-3.1-8B-Instruct** as the target model. For all of these metrics, higher values are better. The best results in each column are marked in bold.

## D.2    GENERALIZATION TO OUT-OF-DISTRIBUTION DATASETS

Extending Table 3, we report the results of in-distribution and out-of-distribution testings for the other rerankers and the other target LLM (Llama-3.1-8B-Instruct) in Table 8. Training on the target dataset yielded both higher compression ratios and stronger QA metrics, while applying the model out of distribution resulted in certain reductions in compression ratios but only minor reductions in the QA metrics (i.e., accuracy and F1).

## D.3    RESULTS USING TOP-10 RETRIEVED DOCUMENTS

We also evaluated the compression methods under the setting where the retrieval step provided the top-10 documents as context, and EFFCOMP processed the top-10 documents directly without running Step 1. Table 9 reports the results on HotpotQA, 2WikiMultiHopQA, Natural Questions, and TriviaQA. The strongest performance is observed with Gemma-2-9B-IT as the target model, likely because our sentence selector was aligned with it during finetuning. Performance with Llama-3.1-8B-Instruct is comparatively lower, yet still surpasses the raw document baseline. This demonstrates that our approach remains beneficial even when the sentence selector is not directly finetuned using signals from the target LLM.

## D.4    SEMANTIC-BASED EVALUATION USING BERTSCORE

Our analysis reveals that traditional metrics based on string matching often mistreat semantically correct predictions as errors. We find that a substantial portion of these "errors" arise from limitations of the string matching metrics rather than true model failures. In contrast, semantic-based metrics such as BERTScore (Zhang et al., 2020) provide a more faithful measure of answer quality. Following this insight, we include BERTScore in our evaluation to provide a more comprehensive assessment of semantic correctness. Specifically, we compute BERT-F1 using contextualized embeddings the microsoft/deberta-xlarge-mnli model (He et al., 2021) at layer 40. Tables 10 and 11

| Method | 2WikiQA | | | NQ | | | TQA | | |
|---|---|---|---|---|---|---|---|---|---|
| | $1/\tau$ | Acc | F1 | $1/\tau$ | Acc | F1 | $1/\tau$ | Acc | F1 |
| **Gemma-2-9B-IT** | | | | | | | | | |
| No reranker (in-distribution) | **16.1x** | **29.5** | **33.4** | **20.1x** | 30.9 | 37.2 | **48.4x** | **78.9** | **76.8** |
| No reranker (out-of-distribution) | 11.7x | 28.9 | 32.5 | 9.8x | **31.1** | **37.3** | 32.8x | 78.9 | 76.7 |
| Jina reranker (in-distribution) | **29.4x** | 25.5 | 29.4 | **34x** | 29.9 | 35.7 | **78.4x** | **80.1** | 76.7 |
| Jina reranker (out-of-distribution) | 19.3x | **25.8** | **29.6** | 18.8x | 29.7 | 35.4 | 50x | 79.6 | 76.3 |
| GTE reranker (in-distribution) | **29.4x** | 25.4 | 29.1 | 33.2x | 30.5 | 36.0 | 64.3x | **80.5** | **77.1** |
| GTE reranker (out-of-distribution) | 20.4x | 25.0 | 28.8 | 19.2x | 29.7 | 35.2 | 54.1x | 79.8 | 76.5 |
| **Llama-3.1-8B-Instruct** | | | | | | | | | |
| No reranker (in-distribution) | **16.1x** | 21.3 | 23.6 | **20.1x** | 27.6 | 33.1 | **48.4x** | 73.5 | 69.0 |
| No reranker (out-of-distribution) | 11.7x | 20.8 | 23.2 | 9.8x | 26.8 | 32.4 | 32.8x | **74.3** | **69.2** |
| BGE reranker (in-distribution) | **30.3x** | 21.5 | 23.8 | 29.1x | 28.5 | 33.1 | 59.5x | 76.6 | 70.3 |
| BGE reranker (out-of-distribution) | 19.4x | 20.3 | 23.0 | 14.1x | 27.5 | 32.5 | 49.6x | 76.1 | 70.2 |
| Jina reranker (in-distribution) | **29.4x** | 21.8 | 24.1 | **34x** | 29.0 | 33.5 | **78.4x** | **77.1** | 70.8 |
| Jina reranker (out-of-distribution) | 19.3x | 21.3 | 23.9 | 18.8x | 28.2 | 33.1 | 50x | 76.2 | 70.1 |
| GTE reranker (in-distribution) | **29.4x** | 21.6 | 23.9 | 33.2x | 29.4 | 34.0 | 64.3x | 77.3 | 70.8 |
| GTE reranker (out-of-distribution) | 20.4x | 20.8 | 23.4 | 19.2x | 28.3 | 33.3 | 54.1x | 77.2 | **70.9** |

Table 8: Cross-dataset generalization of EFFCOMP using **Gemma-2-9B-IT** and **Llama-3.1-8B-Instruct** as target models, evaluated both with and without the document reranker. We report the mean compression ratio $(1/\tau)$, accuracy (Acc), and F1 score. The in-distribution rows correspond to training on each dataset individually, while the out-of-distribution rows correspond to training only on HotpotQA and directly applying the model to other datasets. For all of these metrics, higher values are better. Bold numbers indicate the best performance in each setting.

| Method | HQA | | | 2WikiQA | | | NQ | | | TQA | | |
|---|---|---|---|---|---|---|---|---|---|---|---|---|
| | $1/\tau$ | Acc | F1 | $1/\tau$ | Acc | F1 | $1/\tau$ | Acc | F1 | $1/\tau$ | Acc | F1 |
| **Gemma-2-9B-IT** | | | | | | | | | | | | |
| Raw Documents | 1x | 32.4 | 39.0 | 1x | 22.7 | 26.7 | 1x | **29.5** | **35.3** | 1x | **78.5** | **75.9** |
| LongLLMLingua | 4.8x | 28.5 | 35.5 | 4.7x | 20.7 | 24.5 | 4.8x | 23.3 | 29.2 | 4.9x | 74.9 | 72.7 |
| Recomp-Extractive | 9.7x | 27.3 | 33.4 | 9.7x | 18.3 | 22.7 | 9.8x | 24.3 | 29.4 | 10.3x | 74.6 | 71.6 |
| Recomp-Abstractive | **43.9x** | 27.9 | 34.1 | **42.2x** | 21.0 | 25.1 | **16.1x** | 27.6 | 33.0 | **46.3x** | 73.0 | 70.7 |
| CompAct | 17.3x | 30.7 | 36.6 | 19.5x | 16.9 | 20.4 | 15.5x | 27.6 | 32.7 | 17.2x | 76.4 | 73.3 |
| Ours: EFFCOMP | 10.3x | **33.2** | **39.7** | 9x | **24.1** | **28.2** | 9x | 29.3 | 34.8 | 16.6x | 78.2 | 75.0 |
| **Llama-3.1-8B-Instruct** | | | | | | | | | | | | |
| Raw Documents | 1x | 26.4 | 30.5 | 1x | 18.4 | 21.1 | 1x | 25.9 | 31.4 | 1x | 73.2 | 68.0 |
| LongLLMLingua | 4.8x | 25.6 | 30.7 | 4.7x | 19.1 | 22.0 | 4.8x | 21.9 | 27.4 | 4.9x | 71.3 | 66.8 |
| Recomp-Extractive | 9.7x | 24.8 | 28.7 | 9.7x | 17.7 | 20.7 | 9.8x | 23.5 | 27.7 | 10.3x | 71.2 | 65.2 |
| Recomp-Abstractive | **43.9x** | 27.8 | 32.2 | **42.2x** | 21.2 | 24.5 | **16.1x** | 26.3 | 30.4 | **46.3x** | 74.2 | 68.4 |
| CompAct | 17.3x | 29.7 | 32.7 | 19.5x | 18.3 | 20.5 | 15.5x | **27.5** | 30.7 | 17.2x | **75.1** | 67.2 |
| Ours: EFFCOMP | 10.3x | 28.6 | 32.6 | 9x | 20.6 | 23.3 | 9x | 27.1 | **31.7** | 16.6x | 74.8 | 68.8 |

Table 9: Results on open-domain QA benchmarks with top-10 retrieved documents. We report the mean of compression ratio $(1/\tau)$, accuracy (Acc), and F1 score for **Gemma-2-9B-IT** and **Llama-3.1-8B-Instruct** as target models. For all of these metrics, higher values are better. Bold numbers indicate the best results in each column.

report BERTScore results for both Gemma-2-9B-IT and Llama-3.1-8B-Instruct under the top-30-document and top-10-document settings, respectively. On Gemma-2-9B-IT, EFFCOMP achieves the highest BERTScore on HotpotQA in the top-30 setting and outperforms all other compression methods across datasets. In the top-10 setting, it attains the best BERTScore on HotpotQA and 2WikiMultihopQA and again surpasses all competing compression frameworks across datasets. For Llama-3.1-8B-Instruct, EFFCOMP achieves the highest BERTScore on HotpotQA and Natural Questions in the top-30-document setting and surpasses the raw-document baseline on all datasets. In the top-10-document setting, it obtains the best BERTScore on Natural Questions and exceeds the raw-document baseline on HotpotQA and 2WikiMultihopQA. Overall, the target LLM's predictions under EFFCOMP compression preserve the intended semantics, even though EFFCOMP is not explicitly trained to optimize BERTScore.

| Method | HQA | | 2WikiQA | | NQ | | TQA | |
|---|---|---|---|---|---|---|---|---|
| | $1/\tau$ | BERT(F1) | $1/\tau$ | BERT(F1) | $1/\tau$ | BERT(F1) | $1/\tau$ | BERT(F1) |
| **Gemma-2-9B-IT** | | | | | | | | |
| Raw Documents | 1x | 73.5 | 1x | **69.7** | 1x | **72.7** | 1x | **89.8** |
| BGE | 3x | 72.3 | 3x | 65.3 | 3x | 70.7 | 3x | 88.8 |
| Jina | 3x | 72.1 | 3x | 65.6 | 3x | 71.1 | 3x | 88.8 |
| GTE | 3x | 72.2 | 3x | 65.5 | 3x | 71.0 | 3x | 88.9 |
| LongLLMLingua | 5.1x | 70.6 | 5.1x | 64.4 | 5.1x | 68.3 | 5.1x | 87.8 |
| Recomp-Extractive | 29.2x | 69.1 | 29.3x | 63.6 | 29.8x | 66.8 | 32x | 86.1 |
| Recomp-Abstractive | **132.1x** | 68.8 | **126.9x** | 66.0 | **48.3x** | 69.5 | **139x** | 85.5 |
| CompAct | 48.7x | 67.2 | 54.7x | 58.4 | 44.2x | 67.6 | 50.2x | 86.8 |
| EFFCOMP (No reranker) | 26.8x | **74.0** | 16.1x | 69.5 | 20.1x | 71.7 | 48.4x | 89.0 |
| EFFCOMP (BGE reranker) | 39.5x | 72.6 | 30.3x | 66.7 | 29.1x | 70.4 | 59.5x | 88.6 |
| EFFCOMP (Jina reranker) | 38.5x | 72.3 | 29.4x | 66.8 | 34x | 70.6 | 78.4x | 88.4 |
| EFFCOMP (GTE reranker) | 40.4x | 72.4 | 29.4x | 66.8 | 33.2x | 70.7 | 64.3x | 88.6 |
| **Llama-3.1-8B-Instruct** | | | | | | | | |
| Raw Documents | 1x | 64.7 | 1x | 59.6 | 1x | 67.1 | 1x | 82.8 |
| BGE | 3x | 67.2 | 3x | 61.9 | 3x | 68.2 | 3x | 84.1 |
| Jina | 3x | 67.0 | 3x | 62.2 | 3x | 68.5 | 3x | **84.5** |
| GTE | 3x | 66.9 | 3x | 62.3 | 3x | 68.8 | 3x | 84.2 |
| LongLLMLingua | 5.1x | 66.8 | 5.1x | 62.4 | 5.1x | 67.4 | 5.1x | 84.2 |
| Recomp-Extractive | 29.2x | 65.2 | 29.3x | 62.1 | 29.8x | 65.5 | 32x | 80.0 |
| Recomp-Abstractive | **132.1x** | 66.4 | **126.9x** | **65.4** | **48.3x** | 67.0 | **139x** | 81.5 |
| CompAct | 48.7x | 64.5 | 54.7x | 58.7 | 44.2x | 66.3 | 50.2x | 80.6 |
| EFFCOMP (No reranker) | 26.8x | 67.5 | 16.1x | 63.2 | 20.1x | 68.8 | 48.4x | 83.8 |
| EFFCOMP (BGE reranker) | 39.5x | **67.9** | 30.3x | 63.6 | 29.1x | 68.7 | 59.5x | 83.6 |
| EFFCOMP (Jina reranker) | 38.5x | 67.8 | 29.4x | 63.8 | 34x | 68.8 | 78.4x | 83.5 |
| EFFCOMP (GTE reranker) | 40.4x | 67.8 | 29.4x | 63.8 | 33.2x | **69.1** | 64.3x | 83.3 |

Table 10: Results on **top-30 retrieved documents** for four open-domain QA benchmarks. We report compression ratio $(1/\tau)$ and BERTScore (F1) for **Gemma-2-9B-IT** and **Llama-3.1-8B-Instruct** as the target models. For all of these metrics, higher values are better. The best results in each column are marked in bold.

| Method | HQA | | 2WikiQA | | NQ | | TQA | |
|---|---|---|---|---|---|---|---|---|
| | $1/\tau$ | BERT(F1) | $1/\tau$ | BERT(F1) | $1/\tau$ | BERT(F1) | $1/\tau$ | BERT(F1) |
| **Gemma-2-9B-IT** | | | | | | | | |
| Raw Documents | 1x | 71.0 | 1x | 64.8 | 1x | **70.4** | 1x | **88.3** |
| LongLLMLingua | 4.8x | 69.3 | 4.7x | 63.8 | 4.8x | 67.1 | 4.9x | 87.0 |
| Recomp-Extractive | 9.7x | 69.3 | 9.7x | 63.6 | 9.8x | 67.2 | 10.3x | 86.4 |
| Recomp-Abstractive | **43.9x** | 68.8 | **42.2x** | **66.1** | **16.1x** | 69.5 | **46.3x** | 85.5 |
| CompAct | 17.3x | 67.5 | 19.5x | 59.0 | 15.5x | 67.7 | 17.2x | 86.4 |
| Ours: EFFCOMP | 10.3x | **71.1** | 9x | **66.1** | 9x | 70.0 | 16.6x | 87.9 |
| **Llama-3.1-8B-Instruct** | | | | | | | | |
| Raw Documents | 1x | 65.8 | 1x | 61.9 | 1x | 67.8 | 1x | **83.6** |
| LongLLMLingua | 4.8x | 66.2 | 4.7x | 62.4 | 4.8x | 66.2 | 4.9x | 82.3 |
| Recomp-Extractive | 9.7x | 65.6 | 9.7x | 62.5 | 9.8x | 66.1 | 10.3x | 80.6 |
| Recomp-Abstractive | **43.9x** | 66.9 | **42.2x** | **65.5** | **16.1x** | 67.2 | **46.3x** | 81.5 |
| CompAct | 17.3x | 64.7 | 19.5x | 58.9 | 15.5x | 66.2 | 17.2x | 80.4 |
| Ours: EFFCOMP | 10.3x | 66.7 | 10x | 63.3 | 9x | **68.0** | 16.6x | 83.0 |

Table 11: Results on **top-10 retrieved documents** for four open-domain QA benchmarks. We report compression ratio $(1/\tau)$ and BERTScore (F1) for **Gemma-2-9B-IT** and **Llama-3.1-8B-Instruct** as the target models. For all of these metrics, higher values are better. The best results in each column are marked in bold.

## D.5 ADDITIONAL COMPARISON WITH OTHER COMPRESSION METHODS

To broaden the evaluation, we conduct supplementary experiments comparing our method with three additional compression approaches: CPC (Liskavets et al., 2025), LLMLingua2 (Pan et al.,

| Method | HotpotQA | | | | MuSiQue | | | | SQuAD | | | |
|---|---|---|---|---|---|---|---|---|---|---|---|---|
| | Comp | Acc | F1 | BERT | Comp | Acc | F1 | BERT | Comp | Acc | F1 | BERT |
| Raw documents | 1x | 34.6 | 42.3 | 73.5 | 1x | 8.8 | 14.8 | 63.8 | 1x | 40.3 | 45.3 | **74.2** |
| CPC | 5.6x | 30.5 | 36.9 | 70.4 | 5.3x | 9.2 | 16.0 | 62.8 | 5.1x | 42.4 | 45.5 | 72.7 |
| LongLLMLingua | 5.1x | 31.8 | 38.2 | 70.6 | 5.1x | 9.4 | 15.4 | 61.8 | 5x | 40.1 | 44.3 | 72.3 |
| DAC | 5.1x | 24.7 | 32.3 | 68.1 | 5.1x | 6.4 | 12.9 | 60.2 | 5.3x | 25.4 | 32.4 | 67.2 |
| LLMLingua2 | 5.4x | 25.0 | 32.6 | 68.9 | 5.4x | 7.4 | 14.0 | 63.1 | 5.3x | 26.5 | 31.7 | 67.8 |
| EFFCOMP (No reranker) | 26.8x | **37.3** | **44.2** | **74.0** | 6x | 7.6 | 13.8 | 62.3 | 36.2x | **43.3** | **47.1** | 74.0 |
| EFFCOMP (BGE) | **39.5x** | 36.4 | 43.1 | 72.6 | 9.9x | 8.7 | 15.4 | 63.1 | **42.2x** | 41.3 | 44.6 | 72.4 |
| EFFCOMP (BGE+OOD) | – | – | – | – | **16.8x** | **10.8** | **17.6** | **64.1** | 32.1x | 40.3 | 43.5 | 72.1 |

Table 12: Comparison with CPC, LLMLingua2, and DAC across HotpotQA, MuSiQue, and SQuAD using the top 30 retrieved documents and **Gemma-2-9B-IT** as the target model. Metrics include compression ratio, accuracy, F1 score, and BERTScore.

| Method | Compressor Memory (MB) | LLM Inference Memory (MB) |
|---|---|---|
| Raw documents | – | 11,874.37 |
| LongLLMLingua | 16,129.06 | 6,935.18 |
| Recomp-Extractive | **945.46** | 6,619.45 |
| Recomp-Abstractive | 5,934.80 | 8,208.77 |
| Compact | 14,090.37 | **6,580.15** |
| EFFCOMP (BGE) | 1,252.01 | 7,202.59 |
| EFFCOMP (Jina) | 1,242.26 | 7,141.47 |
| EFFCOMP (GTE) | 1,323.48 | 7,178.40 |

Table 13: Peak GPU memory usage (in MB) for each compressor and the downstream LLM (Gemma-2-9B-IT). Memory is measured during the model forward pass. For Recomp-Abstractive, the peak across 500 examples is reported due to the lack of per-sample measurement. FlashAttention 2 is enabled for models that support it.

2024), and DAC (Zhao et al., 2025a). For CPC, we use the Mistral-7B-Instruct-v0.2 model as in their original paper and set the compression target tokens at 900 to match an approximate 5x compression ratio. For LLMLingua2, we use the microsoft/llmlingua-2-xlm-roberta-large-meetingbank model as the compression model which follows the LLMLingua2 training recipe based on xlm-roberta-large (Conneau et al., 2020), and we set the compression rate at 0.2. For DAC, we use the qwen2-0.5B-instruct model as the compression model with the default configuration, corresponding to a compression ratio of 0.825. All evaluations take the top 30 retrieved documents as input and employ Gemma-2-9B-IT as the target LLM. We report compression ratio, accuracy, F1 score, and BERTScore on HotpotQA, MuSiQue (Trivedi et al., 2022), and SQuAD (Rajpurkar et al., 2016). The results are reported in Table 12. Interestingly, for MuSiQue, our method performs better under the OOD setting (trained on HotpotQA and tested on MuSiQue). This is likely because the fine-tuning set of MuSiQue is very limited (657 samples, compared to over 10,000 samples for the other datasets). Nevertheless, our method remains the top performer across all the compression framework baselines.

### D.6 MEMORY USAGE ANALYSIS

We evaluate the memory footprint of the compressor and the downstream LLM during inference. Memory usage is reported as the average peak GPU memory during the forward pass. For Recomp-Abstractive, per-example measurement is not supported, so we report the peak memory across 500 examples. Table 13 reports the memory consumption of all methods. FlashAttention 2 is enabled for compression models that support it, including LongLLMLingua, Compact, the EFFCOMP (sentence selector), the Jina reranker, and the ModernBERT reranker. Across all methods, EFFCOMP shows competitive memory consumption for both the compression stage and the LLM inference stage.

### D.7 INFORMATION EFFICIENCY ANALYSIS

To analyze how effectively our method removes context-irrelevant information while retaining the essential evidence for answering the question, we evaluate compression performance on the HotpotQA test set. We focus on examples for which the target model predicts the correct answer and

| Method | Compression ratio | Compression ratio (EFFCOMP BGE) |
|---|---|---|
| Recomp-Extractive | 29.11x | 61.41x |
| LongLLMLingua | 5.14x | 61.96x |

Table 14: Compression ratios on correctly answered HotpotQA examples (7,405 total examples). extractive correctly answers 1,657 cases, while LongLLMLingua correctly answers 2,037. Higher values indicate more efficient preservation of information relevant to the QA task.

measure how much each method compresses the context in these successful cases. A higher compression ratio under the same correctness condition indicates that the method preserves the essential information more efficiently.

We compare EFFCOMP with two extractive baselines: Recomp-Extractive and LongLLMLingua. For each method, we compute the average compression ratio on correctly answered examples, as shown in Table 14. Across the dataset, EFFCOMP achieves substantially higher compression ratios than both baselines while answering correctly, indicating that it provides more efficient context-aware selection.

### D.8 ABLATION STUDY ON DIFFERENT FINETUNING STRATEGIES

We conducted an ablation study on the HotpotQA, 2WikiMultiHopQA, Natural Questions, and TriviaQA, evaluating three training strategies: reinforcement learning (RL) only, supervised learning (SL) only with the best trajectory memory, and the hybrid approach of EFFCOMP that combines both. The results are displayed in Figure 4. We observed similar patterns in HotpotQA, Natural Questions, and TriviaQA. Specifically, the hybrid strategy often achieved higher validation accuracy in early epochs, while having a relatively lower compression rate. In contrast, the RL only strategy prioritized the compression rate while sacrificing the accuracy. Meanwhile, the SL only strategy usually achieved higher validation accuracy than the RL one but with a lower and stable compression rate across epochs. For 2WikiMultiHopQA, the trend of the hybrid approach is similar to the other three datasets, whereas the trends of the RL only and SL only strategies are the opposite of those in the other three datasets.

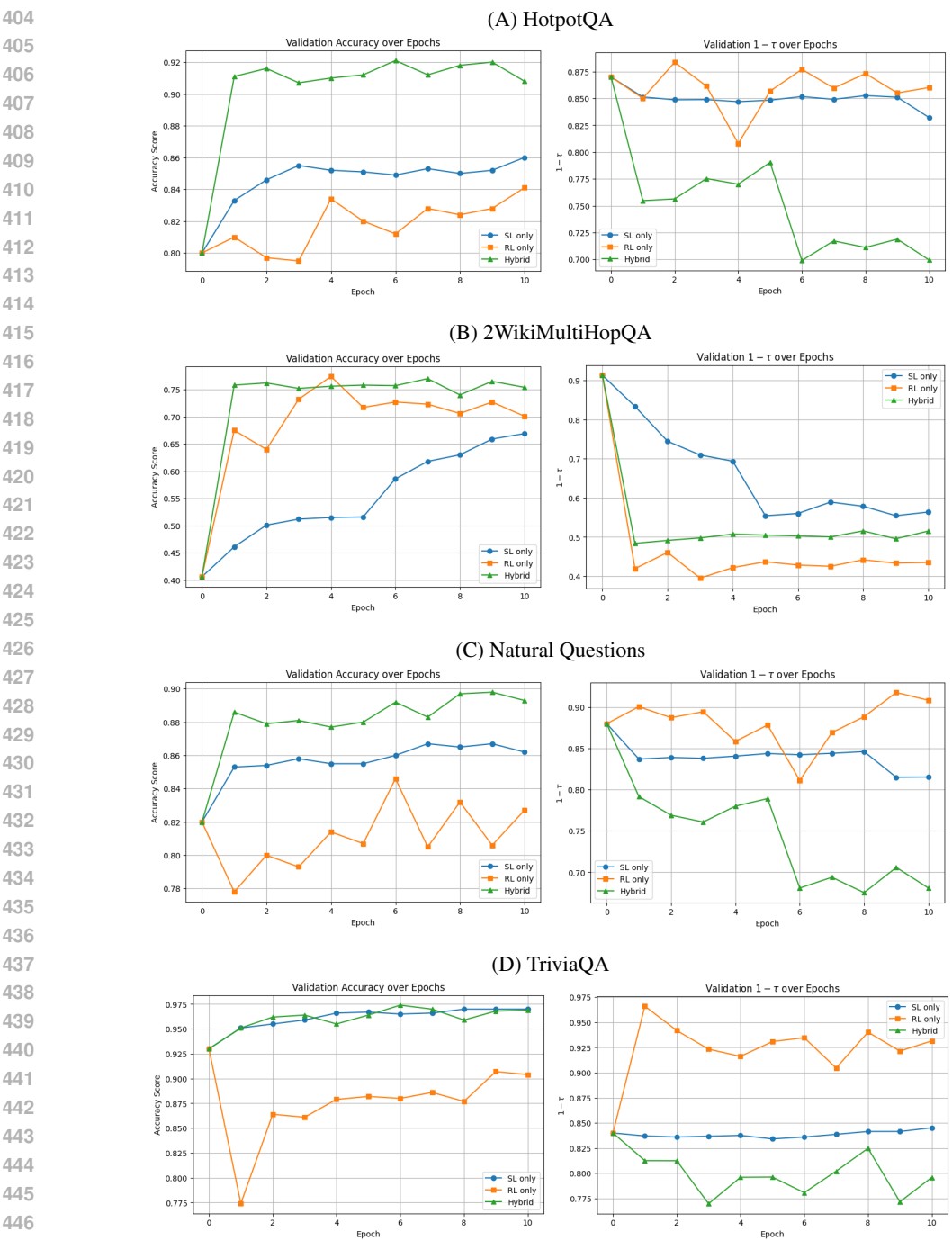

Figure 4: Ablation study of different training strategies for EFFCOMP on the four QA datasets. The plots compare (left) validation accuracy and (right) compression rate $1 - \tau = 1 - |\mathbb{D}|/|\mathbb{C}|$ during validation across 10 epochs. For both metrics, the higher values are better.

## D.9 TRAINING TIME PER EPOCH

Pure RL training is slow because it must repeatedly query the LLM, and relying on RL alone leads to slow convergence in accuracy. In contrast, the SL stage adds only a small overhead while leveraging high-quality trajectories from RL to accelerate learning. As shown in our ablation results in Appendix D.8, the hybrid approach improves accuracy much faster than RL only training.

| Epoch | RL time (sec) | SL time (sec) |
|---|---|---|
| 0 | 58,569.33 | 825.83 |
| 1 | 57,938.17 | 807.11 |
| 2 | 57,715.01 | 840.33 |

Table 15: Per-epoch computation time for the RL and SL components of our training pipeline. The RL time reports the full per-epoch computation cost of the complete RL pipeline, while the SL time reports the additional supervised-learning overhead trained on the trajectories produced by the RL pipeline.

To make this cost difference explicit, Table 15 reports the per-epoch computation times for the full RL stage and the SL overhead. According to the table SL adds only ∼800 seconds per epoch, yet substantially boosts performance when combined with RL. In contrast, RL alone requires ∼58,000 seconds per epoch and converges much more slowly. This makes the hybrid approach both more efficient and more effective.

## D.10 THE EFFECT OF THE PRETRAINING PHASE

This section compares the performance of the sentence selector of EFFCOMP when being finetuned with and without the pretraining phase. We used HotpotQA as the training set and observed the validation accuracy and the compression rate across five epochs. It can be noticed from Figure 5 that the sentence selector without pretraining had lower initial validation accuracy and lower compression rate compared to the pretrained one. Although the compression rates came closer after a few epochs, the validation accuracy of the pretrained sentence selector remained significantly higher than that of the non-pretrained one across the five epochs.

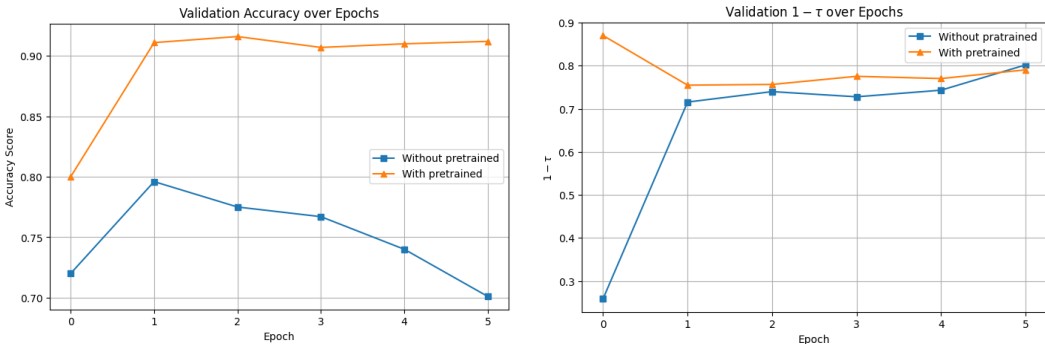

Figure 5: The performance of the sentence selector of EFFCOMP when being finetuned with and without the pretraining phase. The plots show (left) validation accuracy and (right) compression rate $1 - \tau = 1 - |\mathbb{D}|/|\mathbb{C}|$ during validation across five epochs. For both metrics, the higher values are better.

## D.11 IMPACT OF THE WEIGHTING FACTOR $\alpha$

Selecting the best model along the compression–accuracy trade-off requires understanding how different values of $\alpha$ influence the reward signal during validation. We conduct experiments using four values of $\alpha \in \{0.65, 0.75, 0.85, 0.95\}$ for 3 epochs, and evaluate validation rewards on 1,000 samples. Figure 6 illustrates the validation reward, accuracy, and compression rate for each $\alpha$ across the three epochs, where the circled point denotes the checkpoint with the highest reward.

In the validation phase, we observe that for $\alpha = 0.95$, the checkpoint with the highest reward also corresponds to the highest accuracy. For smaller $\alpha$ values, in contrast, the highest reward is primarily achieved by obtaining higher compression rates. This indicates that as $\alpha$ increases, the reward becomes more aligned with accuracy, whereas lower $\alpha$ values encourage more aggressive compression.

| Method | $\alpha$ | Comp. | Acc | F1 |
|---|---|---|---|---|
| EFFCOMP (BGE) | 0.65 | **47.2x** | 35.7 | 42.6 |
| EFFCOMP (BGE) | 0.75 | 33.1x | 36.0 | 42.6 |
| EFFCOMP (BGE) | 0.85 | 46.4x | 36.2 | 42.9 |
| EFFCOMP (BGE) | 0.95 | 39.5x | **36.4** | **43.1** |

Table 16: Test-set HotpotQA (30 retrieved documents) performance using Gemma-2-9B-IT as the target model. We report the performance of the checkpoints selected by the highest validation reward for each $\alpha$. The compression–accuracy behavior on the test set is consistent with that observed during validation.

These results show that $\alpha$ provides a controllable trade-off: larger values prioritize accuracy, while smaller values emphasize compression. This allows users to select checkpoints that best match their desired balance between compression and accuracy within the validation set.

After selecting the highest-reward checkpoints for each $\alpha$ based on validation performance, we evaluate these checkpoints on the HotpotQA test set. The results are shown in Table 16.

Importantly, we observe that the relationship between accuracy and compression on the test set is consistent with the behavior seen during validation. Although our hybrid RL+SL training framework already emphasizes accuracy, using a larger $\alpha$ remains necessary to ensure that model selection follows our context-aware compression objective, where accuracy is the primary criterion.

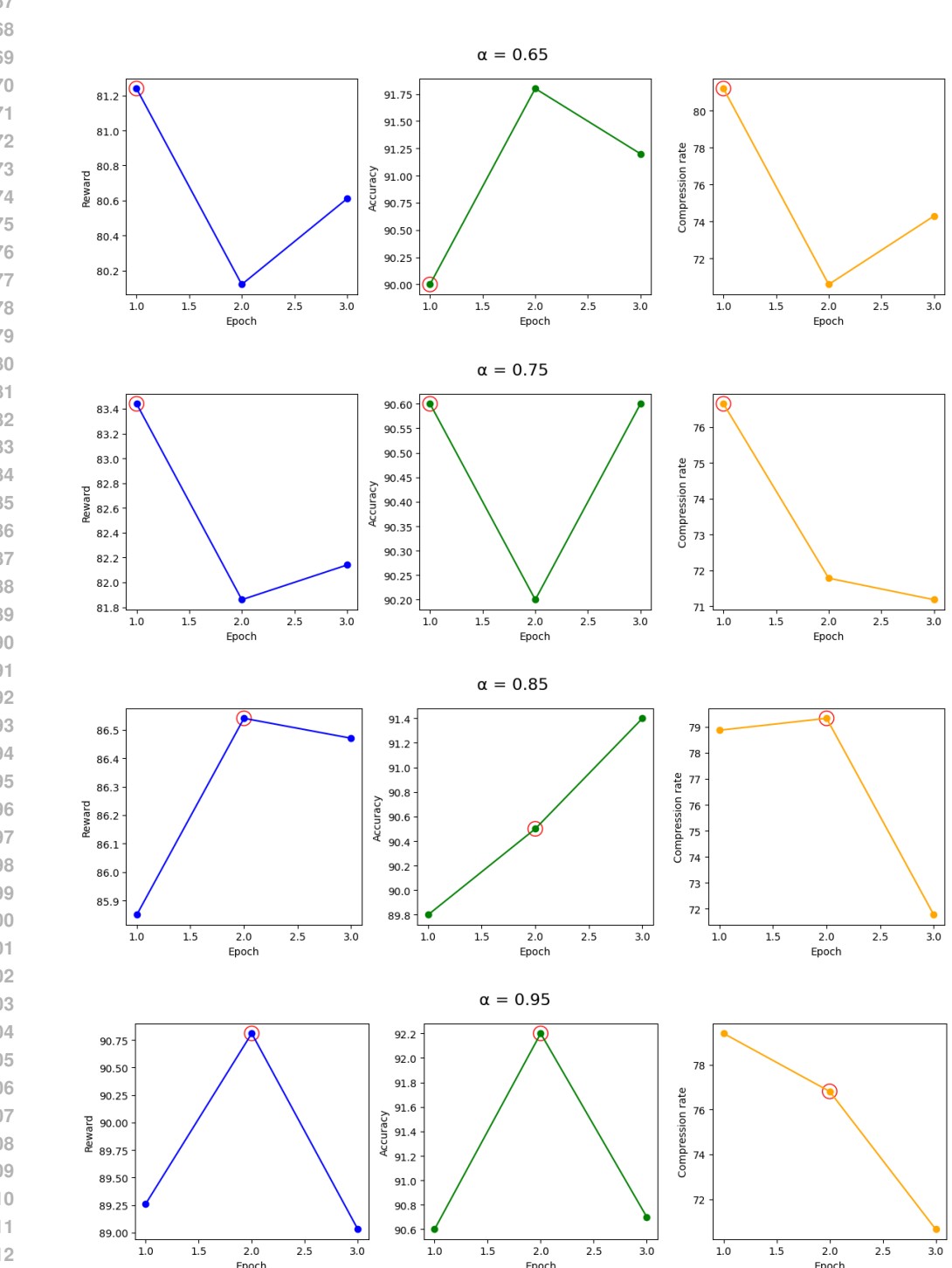

Figure 6: Ablation study of the weighting factor $\alpha \in \{0.65, 0.75, 0.85, 0.95\}$ for EFFCOMP on the HotpotQA validation set. For each value of $\alpha$, we report the Reward, Accuracy, and Compression rate $1 - \tau = 1 - |\mathbb{D}|/|\mathbb{C}|$ across 3 training epochs. The circled point in each plot indicates the checkpoint (epoch) with the highest validation reward. Higher values are better for all metrics.

