# OpenReview forum: "EffComp: Efficient Prompt Compression via Hybrid Reinforcement & Supervised Learning"
_ICLR.cc/2026/Conference — ICLR 2026 Conference Withdrawn Submission_

### Official Review · Reviewer_c8K7 · 2025-10-27

**Soundness:** 3
**Presentation:** 3
**Contribution:** 3
**Rating:** 4
**Confidence:** 4

**Summary:**

The paper introduces EFFCOMP, a hybrid model designed to solve the trade-off in RAG systems by providing high extractive compression while maintaining LLM accuracy and minimizing latency. The main contribution is a novel hybrid of reinforcement and supervised learning for training the sentence selector, optimizing the trade-off between compression ratio and LLM accuracy. EFFCOMP achieves higher QA accuracy through extensive extractive compression and notable reduction in total latency, showing its practical effectiveness.

**Strengths:**

EFFCOMP effectively overcomes the main weakness of previous extractive methods, which only achieved modest compression ratios. It provides significant extractive compression, reaching ratios up to 78.4x while preserving factual accuracy and human readability.

**Weaknesses:**

1. Lack of Quantitative Evidence for Addressing the Core Challenge

>The paper's main goal is to address the inherent trade-off between abstractive compressors (high compression, high hallucination risk) and extractive compressors (low compression, low hallucination risk). The proposed EFFCOMP hybrid method aims to achieve this by offering the advantages of both (high compression and low hallucination). However, the experimental results only report end-to-end QA metrics (Acc/F1) and the compression ratio, failing to quantify the claimed advantage over the competing approaches.

>The paper asserts that EFFCOMP prevents hallucination due to its extractive nature, but this is never quantitatively proven against abstractive methods (like Recomp-Abstractive or CompAct). A rigorous comparison requires quantitatively evaluating their ability to retain factual integrity better than abstractive approaches. The qualitative example in Table 2 is insufficient proof.

>In addition, the paper claims EFFCOMP preserves more important information than other extractive methods, implying high efficiency in selection. This should be demonstrated by showing that the compressed contexts are semantically richer or less redundant than those produced by extractive methods.

>In short, the experiments show that EFFCOMP performs well overall, but they lack the necessary diagnostic analysis to quantitatively validate how it successfully addresses the fundamental trade-off between abstractive hallucination risk and extractive compression limitations, which was the paper's main focus.

2. Lack of Justification for Hybrid RL/SL Architecture

>The paper's core technical contribution is the hybrid Reinforcement Learning (RL) and Supervised Learning (SL) approach used for fine-tuning the sentence selector. While the authors describe the functional purpose of combining the two (RL for optimizing the objective, SL for efficiency via memory), they fail to provide a rigorous justification for why this specific hybrid method is optimal for the task.

>The authors state that the RL loss is needed because initial pseudo-labels are imperfect, and the SL loss (with best-trajectory memory) is needed for efficiency. However, the paper lacks a clear, upfront theoretical or empirical argument explaining why a complex alternating hybrid structure is necessary over simpler, established alternatives.

>Although the paper includes an ablation study (Figure 4, Appendix D.4) comparing RL-only, SL-only, and Hybrid, the comparison is limited to showing which one performs best overall. It does not provide the diagnostic clarity needed to confirm that the benefits outweigh the overhead. Specifically, it doesn't clearly explain why the hybrid method handles zero-reward signals or exploration better than a modified pure-RL structure designed for contextual bandits.

>In essence, the paper proposes a complex solution without fully demonstrating why the problem demands this specific complex solution over more parsimonious alternatives.

**Questions:**

Questions on Latency Metrics and Measurement
> Table 4 provides a detailed breakdown of efficiency, but the units and measurement process need clear explanation to ensure reproducibility. What are the units for the Compression, LLM Read, and Total Latency metrics reported in Table 4? The text suggests they are in seconds, but this should be explicitly stated in the table caption or text. Could the authors provide a concise explanation of how latency was measured?

---

> ### Author Response · Authors · 2025-11-22
> **Response to Reviewer c8K7 (1/3)**
>
> Dear Reviewer,
>
> We greatly appreciate your time and effort in reviewing our paper. Here, we would like to address your concerns point by point:
>
> > **Comment**
> > - The paper asserts that EFFCOMP prevents hallucination due to its extractive nature, but this is never quantitatively proven against abstractive methods (like Recomp-Abstractive or CompAct). A rigorous comparison requires quantitatively evaluating their ability to retain factual integrity better than abstractive approaches. The qualitative example in Table 2 is insufficient proof.
>
> A well-established finding in the summarization literature is that abstractive method is more prone to hallucination than extractive method, since paraphrasing or regenerating text can introduce content not grounded in the source. Evidence for this observation appears broadly across the literature, including a recent survey [1].
>
>
> Nevertheless, to provide quantitative evidence rather than relying solely on prior knowledge, we conducted an additional evaluation comparing our extractive method against two abstractive baselines: RECOMP-Abstractive and CompAct.
>
>
> Experimental setup.
>  We evaluate hallucination using a natural language inference (NLI) task on the HotpotQA test set. For each example, the original raw context is treated as the premise and the compressed context as the hypothesis. We then apply the tasksource/ModernBERT-large-nli (0.4B) model [2] to obtain NLI predictions. Among the output labels, we focus on contradiction, which directly indicates factual inconsistency introduced during compression.
>
>
> What this measures.
> A contradiction prediction indicates that the compressed text contains information conflicting with the original context. Thus, the total number of contradiction cases provides a direct quantitative measure of how often each method introduces factual errors.
>
>
> Below, we report the number of contradiction cases detected on the HotpotQA test set.
>
>
> | Compression Method     | Contradiction |
> |------------------------|---------------|
> | Recomp abstractive     | 1728|
> | Compact| 225|
> | EffComp (BGE Reranker) | **42**|
>
>
> Across all compression methods evaluated on the same dataset, EFFCOMP produces the fewest contradiction cases, demonstrating that our extractive approach preserves factual integrity more reliably than the abstractive compressors. This directly supports our claim that EFFCOMP reduces hallucination risk due to its extractive design.
>
>
> [1] Aisha Alansari el al., A Comprehensive Survey of Hallucination in Large Language Models: Causes, Detection, and Mitigation. Arxiv 2025
>
>
> [2] Damien Sileo., tasksource: A Large Collection of NLP tasks with a Structured Dataset Preprocessing Framework. lrec 2024.

---

> ### Author Response · Authors · 2025-11-22
> **Response to Reviewer c8K7 (2/3)**
>
> > **Comment**
> > - In addition, the paper claims EFFCOMP preserves more important information than other extractive methods, implying high efficiency in selection. This should be demonstrated by showing that the compressed contexts are semantically richer or less redundant than those produced by extractive methods.
>
>
> While the reviewer suggests demonstrating that EFFCOMP produces “semantically richer or less redundant” compressed contexts, we would like to clarify that this property is more aligned with the goals of task-agnostic compression. Task-agnostic compressors aim to reduce redundancy and preserve broad semantic content regardless of the downstream task. In contrast, EFFCOMP is explicitly designed for context-aware compression, where the primary objective is to remove context-irrelevant information while maintaining or improving downstream QA accuracy. Semantic richness is therefore a secondary effect, not the central goal of our method.
>
>
> To provide quantitative evidence of information efficiency in the context-aware setting, we conducted two analyses in HotpotQA test dataset:
>
>
> (1) Compression ratio on correctly answered examples (context-aware efficiency).
>
>
> For instances where the QA prediction is correct, we compare the compression ratios of EFFCOMP with two extractive baselines: ReComp-Extractive and LongLLMLingua. Achieving a higher compression ratio while still answering correctly indicates that the compressed context contains less context-irrelevant content and preserves the necessary information more efficiently.
>
>
> | Method            | Compression ratio | Compression ratio (EffComp BGE) |
> |----|--------|-----|
> |Recomp extractive|29.11x|61.41x|
> |LongLLMLingua|5.14x|61.96x|
>
>
> Across the dataset, EFFCOMP consistently achieves higher compression ratios than both baselines.
>
>
> (2) Semantic relevance between the query and compressed context (context-aware informativeness).
> To further assess how well each compressed context preserves the necessary content, we compute query–context relevance using the reranker model Alibaba-NLP/gte-multilingual-reranker-base (0.3B) [1].
>
>
> | Method  | Relevance Score |
> |-----|--|
> | Base    | **0.84**     |
> | Recomp Extractive| 0.68 |
> | LongLLMLingua  | 0.75  |
> | EffComp(BGE)     | 0.77  |
>
>
> Although all compressed contexts score lower than the raw document, as expected, EFFCOMP achieves higher relevance scores than the other extractive baselines even at higher compression ratios, indicating that it better preserves the information most useful for answering the query.
>
> > **Comment**
> > - The authors state that the RL loss is needed because initial pseudo-labels are imperfect, and the SL loss (with best-trajectory memory) is needed for efficiency. However, the paper lacks a clear, upfront theoretical or empirical argument explaining why a complex alternating hybrid structure is necessary over simpler, established alternatives.
> Although the paper includes an ablation study (Figure 4, Appendix D.4) comparing RL-only, SL-only, and Hybrid, the comparison is limited to showing which one performs best overall. It does not provide the diagnostic clarity needed to confirm that the benefits outweigh the overhead.
>
>
> We agree that providing a clearer motivation for the hybrid structure would strengthen the paper. Prior work supports the need for computational efficiency: Shandilya et al. [2] note that reinforcement-learning based prompt compression requires substantial computational resources, and that improving the efficiency of the fine-tuning process is an important open challenge.
>
> Pure RL training is slow because it must repeatedly query the LLM, while the SL stage serves as a lightweight overhead plug-in that uses high-quality trajectories from the RL phase to reduce optimization cost. As shown in Figure 4, RL-only training converges substantially more slowly than the hybrid approach in terms of accuracy. To further clarify the cost difference, we provide below the actual per-epoch computation times for the full RL pipeline and the small SL overhead.
>
>
> Our ablation results in Figure 4 show that RL-only converges significantly slower than the hybrid approach in terms of accuracy. To further clarify the cost difference, we include below the actual per-epoch computation times for full pipeline RL and overhead SL training:
>
>
> | Epoch | RL time(sec) | SL time(sec) |
> |---|----|---|
> | 0   | 58,569.33   | 825.83    |
> | 1   | 57,938.17    | 807.11    |
> | 2   | 57,715.01    | 840.33    |
>
> SL adds only ~800 seconds per epoch and quickly improves performance. RL takes ~58,000 seconds per epoch, and using RL alone would require far more epochs to reach similar accuracy quality. This motivates adopting a hybrid approach.
>
>
> [1] Zhang et al., mGTE: Generalized Long-Context Text Representation and Reranking Models for Multilingual Text Retrieval. EMNLP industry 2024,
>
>
> [2] Shandilya el al., TACO-RL: Task Aware Prompt Compression Optimization with Reinforcement Learning. ACL Finding 2025.

---

> ### Author Response · Authors · 2025-11-22
> **Response to Reviewer c8K7 (3/3)**
>
> > **Comment**
> > - Specifically, it doesn't clearly explain why the hybrid method handles zero-reward signals or exploration better than a modified pure-RL structure designed for contextual bandits.
>
>
> We appreciate the reviewer’s question. In our setting, issues such as insufficient exploration or zero-reward signals are unlikely, as the reward is dense by design and the policy maintains high entropy. Our method is not intended to address zero-reward cases. Instead, the motivation for the hybrid structure is to make RL training more efficient, which aligns with the practical challenges highlighted in works such as TacoRL.
>
>
> > **Comment**
> > - The paper's core technical contribution is the hybrid Reinforcement Learning (RL) and Supervised Learning (SL) approach used for fine-tuning the sentence selector. While the authors describe the functional purpose of combining the two (RL for optimizing the objective, SL for efficiency via memory), they fail to provide a rigorous justification for why this specific hybrid method is optimal for the task.
>
> In line 173, when we state that “the pretrained model from Phase I is not yet optimal,” the word optimal does not mean globally best. Instead, it reflects the fact that the Phase I model, which is trained using pseudo-labels produced by simple string matching heuristics, has not yet learned to perform compression in the way required for our target setting.
>
>
> To directly address the reviewer’s concern about whether the finetuned model is able to perform the compression task, we conducted an additional experiment on HotpotQA, where multi-hop reasoning depends critically on the preservation of intermediate reasoning sentences (supporting facts). We compared two sentence selectors:
> (1) a pretrained-only compression model using the Phase I objective, and (2) a pretrained + finetuned model using both Phase I and Phase II GRPO. We then evaluated whether the compressed prompts produced by each model retained the annotated supporting facts in the HotpotQA test set. A supporting fact was considered preserved if at least 50% string overlap was detected between the supporting sentence and the compressed text , original document. Our evaluation metric is defined as:
>
>
> Retention Rate = (number of supporting sentences preserved in the compressed text) / (number of supporting sentences in the original document)
>
>
> The pretrained-only model achieved 75.77% retention, whereas the pretrained + finetuned model achieved 91.09%. This demonstrates that Phase II GRPO effectively corrects the initial bias from Phase I and significantly improves the retention of intermediate-reasoning sentences, confirming that the RL stage is capable of exploring and reinforcing these crucial multi-hop reasoning signals.
>
>
> > **Comment**
> > - In essence, the paper proposes a complex solution without fully demonstrating why the problem demands this specific complex solution over more parsimonious alternatives.
>
>
> While the framework may appear complex at first glance due to the nature of the compression task, the core mechanism we employ is actually quite simple. In short, in our hybrid approach, the supervised learning component simply selects the positive highest reward trajectory for each state and performs supervised fine-tuning on these trajectories at every epoch.
>
>
> > **Comment**
> > - Table 4 provides a detailed breakdown of efficiency, but the units and measurement process need clear explanation to ensure reproducibility. What are the units for the Compression, LLM Read, and Total Latency metrics reported in Table 4? The text suggests they are in seconds, but this should be explicitly stated in the table caption or text. Could the authors provide a concise explanation of how latency was measured?
>
>
> We thank the reviewer for raising this point. All latency values in Table 4 are reported in seconds, and we will make this explicit in the table caption in the revised version.
>
> Compression latency is measured by running the full compression pipeline, which includes tokenization, the model’s forward pass, and detokenization, on 500 examples. The total runtime is divided by 500 to obtain the average per-example latency (in seconds). This procedure is applied consistently across all baselines and our method.
> Inference latency is measured using Gemma-2 9B IT with the same procedure. We run tokenization, the forward pass, and detokenization on 500 examples and report the average per example latency.

---

### Official Review · Reviewer_3s4F · 2025-10-28

**Soundness:** 3
**Presentation:** 3
**Contribution:** 3
**Rating:** 6
**Confidence:** 4

**Summary:**

This paper proposes a new sentence-level prompt compression framework for open-domain question answering, which is good for the high compression ratio and low compression latency. To achieve this goal, this paper designs a complex training framework, including supervised learning and end-to-end RL. Then the experiments over four open-domain QA datasets showed the advantages of high compression ratio and accuracy compared with many competitive baselines. This paper also provides many ablation study including the compression latency and fail explanations.

**Strengths:**

This manuscript has many strengths, including:

1. Reasonable method design, including sentence-level, end-to-end RL training
2. This paper conducts comprehensive experiments to show their advantages and provides detailed ablation studies to further investigate.
3. This paper has a good presentation and makes enough contribution to the area of prompt compression.

**Weaknesses:**

1. It's recommended to provide the code to reproduce the results.
2. The main results use F1 metric and EM accuracy. Why doesn't it use the LLM Judge to evaluate the answers?
3. There are some important train-free attention-based methods [1,2] requiring to compare or discussion.

[1] Leveraging Attention to Effectively Compress Prompts for Long-Context LLMs
[2] Efficient Prompt Compression with Evaluator Heads for Long-Context Transformer Inference

**Questions:**

1. In my opinion, the compression ratio of the proposed method is determined adaptively. The compression ratio of selected baselines should be determined by the user. How do you select the compression ratio and what are the reasons?

2. Compression times are dependent on the input prompt. It's better to provide the length of used prompt in Table 4?

---

> ### Author Response · Authors · 2025-11-22
> **Response to Reviewer 3s4F (1/3)**
>
> Dear Reviewer,
>
> We greatly appreciate your time and effort in reviewing our paper. Here, we would like to address your concerns point by point:
>
> > **Comment**
> > - It's recommended to provide the code to reproduce the results.
>
> We appreciate the recommendation, and we will release the code upon paper acceptance.
>
> > **Comment**
> > - The main results use F1 metric and EM accuracy. Why doesn't it use the LLM Judge to evaluate the answers?
>
> Our main results are based on F1 and Accuracy because these metrics are the standard evaluation metrics in prior QA and compression work [1-7], ensuring direct comparability with existing baselines. While LLM-based judges can provide richer semantic feedback, they are significantly more resource-intensive to run at scale, especially across multiple datasets and baselines. For this reason, using an LLM judge for every evaluation example is not practical within our computational constraints. However, following the reviewer’s suggestion for more semantic evaluation, we will additionally report BERTScore in the paper, which offers a lightweight yet semantically informed metric.
>
>
> The QA results for top-30 retrieved documents using Gemma-2-9B-IT, covering compression ratio, accuracy, F1, and BERT F1, are shown below. The revised paper, which will be uploaded soon, will also include the results for Llama-3.1-8B and the top-10 retrieved documents for both models.
>
>
> | Method | HotpotQA |     |     |       | 2WikiQA |     |     |       | NQ |     |     |       | TQA |     |     |       |
> |--------|----------|-----|-----|-------|---------|-----|-----|-------|----|-----|-----|-------|------|-----|-----|-------|
> |        | Comp     | Acc | F1  | BERT  | Comp    | Acc | F1  | BERT  | Comp | Acc | F1 | BERT | Comp | Acc | F1 | BERT |
> | Raw documents | 1x | 34.6 | 42.3 | 73.5 | 1x | 29.1 | 32.9 | **69.7** | 1x | 30.1 | **38.0** | **72.7** | 1x | 77.4 | 77.2 | **89.8** |
> | BGE Reranker | 3x | 35.2 | 42.1 | 72.3 | 3x | 23.8 | 27.6 | 65.3 | 3x | 29.6 | 35.4 | 70.7 | 3x | 79.9 | 77.1 | 88.8 |
> | Jina Reranker | 3x | 34.6 | 41.5 | 72.1 | 3x | 24.4 | 28.2 | 65.6 | 3x | 30.6 | 36.7 | 71.1 | 3x | 80.1 | 77.1 | 88.8 |
> | GTE Reranker | 3x | 34.7 | 41.6 | 72.2 | 3x | 23.9 | 27.5 | 65.5 | 3x | 30.6 | 36.4 | 71.0 | 3x | 80.4 | **77.4** | 88.9 |
> | LongLLMLingua | 5.1x | 31.8 | 38.2 | 70.6 | 5.1x | 21.8 | 25.8 | 64.4 | 5.1x | 25.7 | 31.7 | 68.3 | 5.1x | 78.4 | 75.1 | 87.8 |
> | Recomp-Extractive | 29.2x | 25.6 | 31.9 | 69.1 | 29.3x | 17.4 | 21.4 | 63.6 | 29.8x | 22.8 | 28.2 | 66.8 | 32x | 73.1 | 70.4 | 86.1 |
> | Recomp-Abstractive | **132.1x** | 27.9 | 34.0 | 68.8 | **126.9x** | 21.0 | 25.1 | 66.0 | **48.3x** | 27.6 | 33.1 | 69.5 | **139x** | 73.0 | 70.7 | 85.5 |
> | CompAct | 48.7x | 31.2 | 36.8 | 67.2 | 54.7x | 16.8 | 20.1 | 58.4 | 44.2x | 27.8 | 32.8 | 67.6 | 50.2x | 77.6 | 74.3 | 86.8 |
> | EffComp (No reranker) | 26.8x | **37.4** | **44.3** | **74.0** | 16.1x | **29.5** | **33.4** | 69.5 | 20.1x | **30.9** | 37.2 | 71.7 | 48.4x | 78.9 | 76.8 | 89.0 |
> | EffComp (BGE reranker) | 39.5x | 36.3 | 43.0 | 72.6 | 30.3x | 25.4 | 29.1 | 66.7 | 29.1x | 29.8 | 35.4 | 70.4 | 59.5x | 80.1 | 76.8 | 88.6 |
> | EffComp (Jina reranker) | 38.5x | 35.8 | 42.5 | 72.3 | 29.4x | 25.5 | 29.4 | 66.8 | 34x | 29.9 | 35.7 | 70.6 | 78.4x | 80.1 | 76.7 | 88.4 |
> | EffComp (GTE reranker) | 40.4x | 35.5 | 42.4 | 72.4 | 29.4x | 25.4 | 29.1 | 66.8 | 33.2x | 30.5 | 36.0 | 70.7 | 64.3x | **80.5** | 77.1 | 88.6 |
>
>
> The results show that our method achieves higher BERTScore than using the raw documents on HotpotQA, whereas other compression frameworks do not surpass the raw-document baseline. Notably, even though EFFCOMP is not trained to optimize BERTScore, it still attains the highest BERTScore across all compression methods
>
>
> [1] Zhang, et al., AdaComp: Extractive Context Compression with Adaptive Predictor for Retrieval-Augmented LLMs, Arxiv 2024.
>
>
> [2] Chuang, et al. Learning to Compress Prompt in Natural Language Formats, NAACL 2024.
>
>
> [3] Xu, et al., RECOMP: Improving Retrieval-Augmented LMs with Context Compression and Selective Augmentation, ICLR 2024.
>
>
> [4] Yoon, et al., CompAct: Compressing Retrieved Documents Actively for Question Answering, EMNLP 2024.
>
>
> [5] Jung, el al., Familiarity-Aware Evidence Compression for RAG, EMNLP 2025 Findings.
>
>
> [6] Cheng, el al., xRAG: Extreme Context Compression for Retrieval-Augmented Generation, NeurIPS 2024.
>
>
> [7] Rai, el al., Context Embeddings for Efficient Answer Generation in RAG, WSDM 2025.

---

> ### Author Response · Authors · 2025-11-23
> **Response to Reviewer 3s4F (2/3)**
>
> > **Comment**
> > - There are some important train-free attention-based methods [1,2] requiring to compare or discussion.
>
> We thank the reviewer for pointing out these important training-free, attention-based compression methods. Unfortunately, we could not find publicly available code or reproducible implementations for [1]. For [2], the work is very recent (released <2 months ago), and the released version does not support Gemma-2-9B-IT, which prevents a fair comparison in our main experiments. In addition, their framework embeds both the task instruction and the context into the compressed prompt, making the compression ratio difficult to compare directly. For these reasons, we were unable to include [1–2] in our empirical evaluation.
>
>
> We will add both methods to the Related Work section of the revised paper. The updated version will be uploaded soon. Meanwhile, we have also evaluated recent 2025 methods, including DAC [3] and CPC [4], using the dataset recommended by reviewer xN74.
>
>
> Results of Compression Framework Evaluation Using Gemma-2-9B-IT on 30 documents
>
> | Method | HotpotQA |     |     |      | Musique |     |     |      | SQuAD |     |     |      |
> |-----|-----|-----|-----|-----|-------|-----|----|------|-------|-----|---|----|
> |        | Comp     | Acc | F1  | BERT | Comp    | Acc | F1  | BERT | Comp  | Acc | F1  | BERT |
> | Raw documents | 1x | 34.6 | 42.3 | 73.5 | 1x | 8.8 | 14.8 | 63.8 | 1x | 40.3 | 45.3 | **74.2** |
> | CPC | 5.6x | 30.5 | 36.9 | 70.4 | 5.3x | 9.2 | 16.0 | 62.8 | 5.1x | 42.4 | 45.5 | 72.7 |
> | LongLLMLingua | 5.1x | 31.8 | 38.2 | 70.6 | 5.1x | 9.4 | 15.4 | 61.8 | 5x | 40.1 | 44.3 | 72.3 |
> | DAC | 5.1x | 24.7 | 32.3 | 68.1 | 5.1x | 6.4 | 12.9 | 60.2 | 5.3x | 25.4 | 32.4 | 67.2 |
> | LLMLingua2 | 5.4x | 25.0 | 32.6 | 68.9 | 5.4x | 7.4 | 14.0 | 63.1 | 5.3x | 26.5 | 31.7 | 67.8 |
> | Our method (No reranker) | 26.8x | **37.3** | **44.2** | **74.0** | 6.0x | 7.6 | 13.8 | 62.3 | 36.2x | **43.3** | **47.1** | 74.0 |
> | Our method (BGE) | **39.5x** | 36.4 | 43.1 | 72.6 | 9.9x | 8.7 | 15.4 | 63.1 | **42.2x** | 41.3 | 44.6 | 72.4 |
> | Our method (BGE+OOD) | - | - | - | - | **16.8x** | **10.8** | **17.6** | **64.1** | 32.1x | 40.3 | 43.5 | 72.1 |
>
> Interestingly, for MuSiQue, our method performs better under the OOD setting (trained on HotpotQA and tested on MuSiQue). This is likely because the fine-tuning set of MuSiQue is very limited (657 samples, compared to over 10,000 samples for the other datasets). Nevertheless, our method remains the top performer across all compression framework baselines.
>
> > **Comment**
> > - In my opinion, the compression ratio of the proposed method is determined adaptively. The compression ratio of selected baselines should be determined by the user. How do you select the compression ratio and what are the reasons?
>
> Our method determines the compression ratio adaptively through the learned policy, whereas baseline methods require the compression ratio to be manually specified. To ensure fairness, we follow the configurations commonly used or recommended in their original papers:
> - RECOMP (extractive).
> RECOMP operates at the sentence level, the same granularity as our method. Prior work by Yoon et al [5] evaluates RECOMP using the top 4 extracted sentences. We adopt this setting because it matches our sentence-level setup and yields a comparable compression ratio.
> - LongLLMLingua.
> The original paper recommends a compression range of 2×–6×, so we use a 5× setting, which lies near the upper end of their suggested range. In contrast, our method typically achieves much higher compression ratios (often 20× or more), making the 5× setting a fair and appropriate configuration for evaluating LongLLMLingua.
>
>
> Using Gemma-2-9B-IT on 30 documents from the HotpotQA test set, we observe that LongLLMLingua’s performance decreases noticeably as the compression ratio increases:
> | Method        | Comp  | Acc  | F1   |
> |---|----|-----|-----|
> | LongLLMLingua   | 5.1×     | 31.8  | 38.2 |
> | LongLLMLingua   | 19.1×    | 28.7  | 35.7 |
> As expected, if LongLLMLingua is forced to compress more aggressively, its performance drops substantially.
>
> These choices ensure that each baseline is evaluated under conditions consistent with its intended usage and prior evaluations, allowing for a fair and meaningful comparison with our adaptive compression method.
>
>
> [1] Zhao et al,. Leveraging Attention to Effectively Compress Prompts for Long-Context LLMs. AAAI 2025.
>
>
> [2] Fei et al,. Efficient Prompt Compression with Evaluator Heads for Long-Context Transformer Inference. NeurIPS 2025.
>
>
> [3] Yi Zhao, et al., DAC: A Dynamic Attention-aware Approach for Task-Agnostic Prompt Compression, ACL 2025.
>
>
> [4] Liskavets, et al., Prompt Compression with Context-Aware Sentence Encoding for Fast and Improved LLM Inference, AAAI 2025.
>
>
> [5] Yoon, et al., CompAct: Compressing Retrieved Documents Actively for Question Answering, EMNLP 2024

---

> ### Author Response · Authors · 2025-11-23
> **Response to Reviewer 3s4F (3/3)**
>
> > **Comment**
> > - Compression times are dependent on the input prompt. It's better to provide the length of used prompt in Table 4?
>
>
> Thank you for your comment. After revisiting our evaluation code, we identified an issue that caused the compression and generation phases to run longer than intended for all methods. We have corrected this issue and re-run the experiments with FlashAttention-enabled baselines as recommended by Reviewer xN74. The updated results, together with the corresponding input prompt lengths, will be included in Table 4 of the revised paper to allow readers to more accurately interpret the relationship between input size and compression latency across different methods.
>
>
> | Method             | Compressor Latency (s) | LLM Inference Latency (s) | Total Latency (s) | Avg Prompt Length |
> | ------------------ | ---------------------- | ------------------------- | ----------------- | ----------------- |
> | Raw documents      | –                      | 1.84                      | 1.84              | 3706.81           |
> | LongLLMLingua      | 2.31                   | 0.90                      | 3.21              | 723.69            |
> | Recomp Extractive  | 0.19                   | 0.74                      | 0.93              | 141.71            |
> | Recomp Abstractive | 2.84                   | **0.71**                      | 3.55              | 106.62            |
> | Compact            | 16.30                  | 0.72                      | 17.02             | **89.93**             |
> | EFFCOMP (BGE)      | 0.08                   | 0.82                      | 0.90              | 487.72            |
> | EFFCOMP (Jina)     | **0.04**                   | 0.82                      | **0.86**              | 499.14            |
> | EFFCOMP (GTE)      | 0.12                   | 0.82                      | 0.94              | 482.72            |
>
>
> With the corrected setup, our updated results show that EFFCOMP achieves the lowest total latency among all compression frameworks. The LLM inference latency is slightly higher because the average prompt is longer. This likely occurs because our training strategy encourages the model to compress aggressively when confident, but to retain more content when the input is harder to compress. Even so, the compressor is fast enough to fully offset the additional inference time, making EFFCOMP the fastest method overall under this experimental configuration.

---

> > ### Comment · Reviewer_3s4F · 2025-11-26
> >
> > Thank you for the detailed response, and my concerns have been addressed. I am pleased to support this paper to be accepted because of the effectiveness of the extreme compression ratio.

---

> > > ### Author Response · Authors · 2025-11-26
> > > **Glad to Hear Your Positive Assessment**
> > >
> > > Thank you for taking the time to review our response. We sincerely appreciate your supportive assessment of our work. If possible, we would be grateful if you could consider adjusting your rating so that it is better reflected in the paper’s overall average score.

---

### Official Review · Reviewer_isCC · 2025-10-28

**Soundness:** 3
**Presentation:** 2
**Contribution:** 2
**Rating:** 2
**Confidence:** 3

**Summary:**

This paper introduces EffComp, a two-step, extractive prompt compression framework designed for RAG-based QA.

**Strengths:**

- The framework achieves high compression ratios and outperforms baselines in QA accuracy and F1 score across four datasets.

- The paper includes generalization tests to an out-of-distribution LLM (Llama-3.1-8B) and a failure analysis.

**Weaknesses:**

- The baselines are all from 2024, and the paper lacks a comparison with newer methods from 2025 (such as CoLoR[1] and DAC[2]).

- While other prompt compression papers often demonstrate their method's effectiveness on summarization in addition to QA, this paper's evaluation is confined solely to QA.

- The authors state that 42% of the failure points are attributed to the evaluation metric and suggests that semantic-based evaluation metrics like BERTScore might be a better option. Therefore, the paper should have included evaluations using these metrics.

- Similarly, such an unreliable metric is used as the core signal for the GRPO reward function. This may lead to unstable training and spurious policy updates that do not reflect true semantic quality.

[1] Minju Seo, et al., Efficient Long Context Language Model Retrieval with Compression, ACL 2025.

[2] Yi Zhao, et al., DAC: A Dynamic Attention-aware Approach for Task-Agnostic Prompt Compression, ACL 2025.

**Questions:**

- The Phase I pretraining biases the policy against intermediate reasoning sentences by exclusively labeling sentences containing the final answer string. This creates a low probability for retaining crucial intermediate-reasoning sentences in multi-hop QA (like HotpotQA). It is questionable whether the GRPO sampling in Phase II can effectively explore and reinforce the retention of these sentences, or if it will struggle to overcome the flawed bias established during pretraining.

- The ablation study on RL and SL in the appendix should be placed in the main text, or at least be mentioned.

---

> ### Author Response · Authors · 2025-11-22
> **Response to Reviewer isCC (1/3)**
>
> Dear Reviewer,
>
> We greatly appreciate your time and effort in reviewing our paper. Here, we would like to address your concerns point by point:
>
> > **Comment**
> > - The baselines are all from 2024, and the paper lacks a comparison with newer methods from 2025 (such as CoLoR[1] and DAC[2]).
>
> Our evaluation focuses primarily on context-aware, hard prompt compression methods, as this setting is most aligned with the design goals of EFFCOMP. In general, task-agnostic compressors (e.g., DAC[2]) tend to yield lower QA accuracy compared to context-aware approaches, which is why our initial set of baselines emphasized methods such as LongLLMLingua, Recomp, and CompAct. Nevertheless, following the reviewer’s suggestion (Reviewer xN74), we have updated our experiments to include the 2025 baselines DAC and CPC [3], evaluated on HotpotQA, MuSiQue, and SQuAD.
>
> (We did not include CoLoR[1] in our evaluation because its objective differs substantially from ours. CoLoR is designed for information retrieval, compressing documents to improve retrieval scoring, whereas our work focuses on context compression for downstream QA. Since the task formulation and evaluation pipeline diverge considerably, we believe CoLoR is not a directly comparable baseline. In addition, we were unable to locate publicly available code or reproducible implementations for CoLoR.)
>
> Results of Compression Framework Evaluation Using Gemma-2-9B-IT on 30 documents
>
> | Method | HotpotQA |     |     |      | Musique |     |     |      | SQuAD |     |     |      |
> |--------|----------|-----|-----|------|---------|-----|-----|------|-------|-----|-----|------|
> |        | Comp     | Acc | F1  | BERT | Comp    | Acc | F1  | BERT | Comp  | Acc | F1  | BERT |
> | Raw documents | 1x | 34.6 | 42.3 | 73.5 | 1x | 8.8 | 14.8 | 63.8 | 1x | 40.3 | 45.3 | **74.2** |
> | CPC | 5.6x | 30.5 | 36.9 | 70.4 | 5.3x | 9.2 | 16.0 | 62.8 | 5.1x | 42.4 | 45.5 | 72.7 |
> | LongLLMLingua | 5.1x | 31.8 | 38.2 | 70.6 | 5.1x | 9.4 | 15.4 | 61.8 | 5x | 40.1 | 44.3 | 72.3 |
> | DAC | 5.1x | 24.7 | 32.3 | 68.1 | 5.1x | 6.4 | 12.9 | 60.2 | 5.3x | 25.4 | 32.4 | 67.2 |
> | LLMLingua2 | 5.4x | 25.0 | 32.6 | 68.9 | 5.4x | 7.4 | 14.0 | 63.1 | 5.3x | 26.5 | 31.7 | 67.8 |
> | Our method (No reranker) | 26.8x | **37.3** | **44.2** | **74.0** | 6.0x | 7.6 | 13.8 | 62.3 | 36.2x | **43.3** | **47.1** | 74.0 |
> | Our method (BGE) | **39.5x** | 36.4 | 43.1 | 72.6 | 9.9x | 8.7 | 15.4 | 63.1 | **42.2x** | 41.3 | 44.6 | 72.4 |
> | Our method (BGE+OOD) | - | - | - | - | **16.8x** | **10.8** | **17.6** | **64.1** | 32.1x | 40.3 | 43.5 | 72.1 |
>
> Interestingly, for MuSiQue, our method performs better under the OOD setting (trained on HotpotQA and tested on MuSiQue). This is likely because the fine-tuning set of MuSiQue is very limited (657 samples, compared to over 10,000 samples for the other datasets). Nevertheless, our method remains the top performer across all compression framework baselines.
>
>
>
> > **Comment**
> > - While other prompt compression papers often demonstrate their method's effectiveness on summarization in addition to QA, this paper's evaluation is confined solely to QA.
>
>
> We agree that many prompt compression papers include summarization as an auxiliary task. However, our focus is intentionally limited to QA compression, which aligns with the dominant use case in recent literature. Several influential works design and evaluate their compression methods specifically for retrieval-augmented QA, rather than for general task. Examples [4-10]
> These methods demonstrate that QA-centric compression is a well-established and meaningful evaluation setting. Since our work aims to improve compression specifically for QA tasks, we evaluate our method solely on QA benchmarks, consistent with the scope and motivation of the paper.
>
>
> [1] Minju Seo, et al., Efficient Long Context Language Model Retrieval with Compression, ACL 2025.
>
>
> [2] Yi Zhao, et al., DAC: A Dynamic Attention-aware Approach for Task-Agnostic Prompt Compression, ACL 2025.
>
>
> [3] Liskavets, et al., Prompt Compression with Context-Aware Sentence Encoding for Fast and Improved LLM Inference, AAAI 2025.
>
>
> [4] Zhang, et al., AdaComp: Extractive Context Compression with Adaptive Predictor for Retrieval-Augmented LLMs, Arxiv 2024.
>
>
> [5] Chuang, et al., Learning to Compress Prompt in Natural Language Formats, NAACL 2024.
>
>
> [6] Xu, et al., RECOMP: Improving Retrieval-Augmented LMs with Context Compression and Selective Augmentation, ICLR 2024.
>
>
> [7] Yoon, et al., CompAct: Compressing Retrieved Documents Actively for Question Answering, EMNLP 2024.
>
>
> [8] Jung, el al., Familiarity-Aware Evidence Compression for RAG, EMNLP 2025 Findings.
>
>
> [9] Cheng, el al., xRAG: Extreme Context Compression for Retrieval-Augmented Generation, NeurIPS 2024.
>
>
> [10] Rai, el al., Context Embeddings for Efficient Answer Generation in RAG, WSDM 2025.

---

> ### Author Response · Authors · 2025-11-22
> **Response to Reviewer isCC (2/3)**
>
> > **Comment**
> > - The authors state that 42% of the failure points are attributed to the evaluation metric and suggests that semantic-based evaluation metrics like BERTScore might be a better option. Therefore, the paper should have included evaluations using these metrics.
>
>
> We appreciate the reviewer’s observation. We use accuracy and token overlap based metrics because they are the standard evaluation protocol in prior work [1-7] on compression-based QA and RAG systems, and most benchmarks report results in terms of EM, F1, or accuracy. However, our own analysis reveals that these metrics frequently misclassify semantically correct answers. In our study, we found that a substantial portion of “errors” arise from metric limitations rather than from actual model failures, and we explicitly call out this issue in the paper.
>
>
> As the reviewer suggests, semantic-based metrics such as BERTScore offer a more faithful measure of answer quality. Following this recommendation, we will add BERTScore in the revised paper to offer a more comprehensive assessment of semantic correctness.
>
>
> The QA results for top-30 retrieved documents using Gemma-2-9B-IT, covering compression ratio, accuracy, F1, and BERT F1, are shown below. The revised paper, which will be uploaded soon, will also include the results for Llama-3.1-8B and the top-10 retrieved documents for both models.
>
>
> | Method | HotpotQA |     |     |       | 2WikiQA |     |     |       | NQ |     |     |       | TQA |     |     |       |
> |--------|----------|-----|-----|-------|---------|-----|-----|-------|----|-----|-----|-------|------|-----|-----|-------|
> |        | Comp     | Acc | F1  | BERT  | Comp    | Acc | F1  | BERT  | Comp | Acc | F1 | BERT | Comp | Acc | F1 | BERT |
> | Raw documents | 1x | 34.6 | 42.3 | 73.5 | 1x | 29.1 | 32.9 | **69.7** | 1x | 30.1 | **38.0** | **72.7** | 1x | 77.4 | 77.2 | **89.8** |
> | BGE Reranker | 3x | 35.2 | 42.1 | 72.3 | 3x | 23.8 | 27.6 | 65.3 | 3x | 29.6 | 35.4 | 70.7 | 3x | 79.9 | 77.1 | 88.8 |
> | Jina Reranker | 3x | 34.6 | 41.5 | 72.1 | 3x | 24.4 | 28.2 | 65.6 | 3x | 30.6 | 36.7 | 71.1 | 3x | 80.1 | 77.1 | 88.8 |
> | GTE Reranker | 3x | 34.7 | 41.6 | 72.2 | 3x | 23.9 | 27.5 | 65.5 | 3x | 30.6 | 36.4 | 71.0 | 3x | 80.4 | **77.4** | 88.9 |
> | LongLLMLingua | 5.1x | 31.8 | 38.2 | 70.6 | 5.1x | 21.8 | 25.8 | 64.4 | 5.1x | 25.7 | 31.7 | 68.3 | 5.1x | 78.4 | 75.1 | 87.8 |
> | Recomp-Extractive | 29.2x | 25.6 | 31.9 | 69.1 | 29.3x | 17.4 | 21.4 | 63.6 | 29.8x | 22.8 | 28.2 | 66.8 | 32x | 73.1 | 70.4 | 86.1 |
> | Recomp-Abstractive | **132.1x** | 27.9 | 34.0 | 68.8 | **126.9x** | 21.0 | 25.1 | 66.0 | **48.3x** | 27.6 | 33.1 | 69.5 | **139x** | 73.0 | 70.7 | 85.5 |
> | CompAct | 48.7x | 31.2 | 36.8 | 67.2 | 54.7x | 16.8 | 20.1 | 58.4 | 44.2x | 27.8 | 32.8 | 67.6 | 50.2x | 77.6 | 74.3 | 86.8 |
> | EffComp (No reranker) | 26.8x | **37.4** | **44.3** | **74.0** | 16.1x | **29.5** | **33.4** | 69.5 | 20.1x | **30.9** | 37.2 | 71.7 | 48.4x | 78.9 | 76.8 | 89.0 |
> | EffComp (BGE reranker) | 39.5x | 36.3 | 43.0 | 72.6 | 30.3x | 25.4 | 29.1 | 66.7 | 29.1x | 29.8 | 35.4 | 70.4 | 59.5x | 80.1 | 76.8 | 88.6 |
> | EffComp (Jina reranker) | 38.5x | 35.8 | 42.5 | 72.3 | 29.4x | 25.5 | 29.4 | 66.8 | 34x | 29.9 | 35.7 | 70.6 | 78.4x | 80.1 | 76.7 | 88.4 |
> | EffComp (GTE reranker) | 40.4x | 35.5 | 42.4 | 72.4 | 29.4x | 25.4 | 29.1 | 66.8 | 33.2x | 30.5 | 36.0 | 70.7 | 64.3x | **80.5** | 77.1 | 88.6 |
>
>
> The results show that our method achieves higher BERTScore than using the raw documents on HotpotQA, whereas other compression frameworks do not surpass the raw-document baseline. Notably, even though EFFCOMP is not trained to optimize BERTScore, it still attains the highest BERTScore across all compression methods
>
>
> [1] Zhang, et al., AdaComp: Extractive Context Compression with Adaptive Predictor for Retrieval-Augmented LLMs, Arxiv 2024.
>
>
> [2] Chuang, et al. Learning to Compress Prompt in Natural Language Formats, NAACL 2024.
>
>
> [3] Xu, et al., RECOMP: Improving Retrieval-Augmented LMs with Context Compression and Selective Augmentation, ICLR 2024.
>
>
> [4] Yoon, et al., CompAct: Compressing Retrieved Documents Actively for Question Answering, EMNLP 2024.
>
>
> [5] Jung, el al., Familiarity-Aware Evidence Compression for RAG, EMNLP 2025 Findings.
>
>
> [6] Cheng, el al., xRAG: Extreme Context Compression for Retrieval-Augmented Generation, NeurIPS 2024.
>
>
> [7] Rai, el al., Context Embeddings for Efficient Answer Generation in RAG, WSDM 2025.

---

> ### Author Response · Authors · 2025-11-22
> **Response to Reviewer isCC (3/3)**
>
> > **Comment**
> > - Similarly, such an unreliable metric is used as the core signal for the GRPO reward function. This may lead to unstable training and spurious policy updates that do not reflect true semantic quality.
>
> We acknowledge the reviewer’s concern that using a string-matching based metric as the core signal in the GRPO reward may introduce instability or spurious updates. However, such reward designs remain standard practice in reinforcement learning for QA and reasoning tasks. For example, SuperRL [1] also relies on discrete string-matching correctness signals as its primary reward.
>
>
> In our setting, the reward function uses a binary correctness indicator (correct vs. incorrect) to determine a compression decision. Continuous semantic metrics such as BERTScore cannot replace this reward signal because they lack the discrete decision boundary required for compression policy learning.
>
>
> Moreover, using an LLM-as-a-judge signal is significantly more resource-intensive, and prior work has shown that it does not necessarily offer advantages in QA reward design. Wang et al. [2] provide an ablation study demonstrating that, on QA datasets, exact-match (EM) based rewards outperform LLM as judge rewards for guiding RL optimization.
>
>
> > **Comment**
> > - The Phase I pretraining biases the policy against intermediate reasoning sentences by exclusively labeling sentences containing the final answer string. This creates a low probability for retaining crucial intermediate-reasoning sentences in multi-hop QA (like HotpotQA). It is questionable whether the GRPO sampling in Phase II can effectively explore and reinforce the retention of these sentences, or if it will struggle to overcome the flawed bias established during pretraining.
>
>
> To address the reviewer’s concern regarding whether Phase II GRPO can overcome the bias introduced in Phase I pretraining, we conducted an additional experiment on HotpotQA, where intermediate reasoning sentences (supporting facts) are essential. We trained two models: (1) a pretrained-only compression model using the Phase I objective, and (2) a pretrained + finetuned model using both Phase I and Phase II GRPO. We then evaluated whether the compressed prompts produced by each model retained the annotated supporting facts in the HotpotQA test set. A supporting fact was considered preserved if at least 50% string overlap was detected between the supporting sentence and the compressed text , original document. Our evaluation metric is defined as:
>
>
> Retention Rate = (number of supporting sentences preserved in the compressed text) / (number of supporting sentences in the original document)
>
>
> The pretrained-only model achieved 75.77% retention, whereas the pretrained + finetuned model achieved 91.09%. This demonstrates that Phase II GRPO effectively corrects the initial bias from Phase I and significantly improves the retention of intermediate-reasoning sentences, confirming that the RL stage is capable of exploring and reinforcing these crucial multi-hop reasoning signals.
>
> > **Comment**
> > - The ablation study on RL and SL in the appendix should be placed in the main text, or at least be mentioned.
>
> We agree with the reviewer that the ablation study on the roles of RL and supervised learning (SL) is important for understanding our method’s contribution. We will follow this recommendation by moving the ablation results into the main paper (or explicitly referencing them in the main text) to ensure their significance is clearly highlighted. This change will appear in the revised paper that will be uploaded soon.
>
>
>
> [1] Liu et al., SuperRL: Reinforcement Learning with Supervision to Boost Language Model Reasoning, CoRR, 2025.
>
>
> [2] Wang et al., LoongRL: Reinforcement Learning for Advanced Reasoning over Long Contexts, arXiv, 2025.

---

> > ### Comment · Reviewer_isCC · 2025-11-27
> >
> > I appreciate the authors’ efforts in adding additional experiments. I will raise my score; however, I still lean toward rejection, as the work remains limited to the QA task and is not comparable to existing general prompt compression methods.

---

> > > ### Author Response · Authors · 2025-11-28
> > > **Response to Reviewer isCC**
> > >
> > > We appreciate the reviewer’s comments and the opportunity to clarify our contributions. Our work addresses RAG-based QA prompt compression, a setting where multiple retrieved documents must be incorporated into the prompt, resulting in substantially longer inputs than standard QA. Given the widespread use of RAG systems, we believe this represents a strong and practically relevant motivation.
> > >
> > > Regarding comparability, many existing prompt-compression baselines are also task-specific rather than fully general. Recomp and CompAct, for example, are designed exclusively for QA, while methods such as LongLLMLingua, DAC, CPC, and LLMLingua2 focus on general purpose compression and may not be directly aligned with RAG-QA scenarios [1].
> > >
> > > To ensure fairness, our paper evaluates all relevant baselines across both categories: QA-only methods (Recomp extractive/abstractive, CompAact) and general purpose methods (LongLLMLingua, DAC, CPC, LLMLingua2). Under this unified evaluation, EFFCOMP consistently provides the strongest balance between answer accuracy and compression ratio, which are the metrics most relevant to real-world RAG deployments operating under context-length and latency constraints.
> > >
> > > Although our primary motivation is QA-centric RAG compression, EFFCOMP can be adapted to summarization by framing summarization requests in QA form (e.g., “Question: Summarize this report…”), though such extensions fall outside the intended scope of the current work.
> > >
> > > In summary, EFFCOMP directly tackles the practical challenge of compressing long RAG prompts, aligns with and extends prior QA-focused compression research, and demonstrates clear empirical advantages across all compared algorithms. We hope this clarification helps convey the motivation and significance of our contribution, and we sincerely thank the reviewer for their thoughtful feedback.
> > >
> > > [1] Li et al., Prompt Compression for Large Language Models: A Survey. NAACL 2025

---

### Official Review · Reviewer_xN74 · 2025-11-01

**Soundness:** 2
**Presentation:** 3
**Contribution:** 2
**Rating:** 4
**Confidence:** 3

**Summary:**

This paper introduces EFFCOMP, an efficient prompt compression framework for RAG-based open-domain question answering. The method employs a two-stage approach: (1) document-level reranking using BERT-style rerankers to filter and reorder retrieved documents, and (2) sentence-level selection using a BERT-based sentence selector trained through a hybrid reinforcement and supervised learning approach. The sentence selector is first pretrained using pseudo-labels based on answer string matching, then finetuned using GRPO with a reward function that balances QA accuracy and compression ratio. During finetuning, a best trajectory memory stores high-reward action sequences for supervised learning updates. Experiments on four QA datasets (Natural Questions, TriviaQA, HotpotQA, 2WikiMultiHopQA) with Gemma-2-9B-IT demonstrate compression ratios of 29.1x-78.4x while maintaining or improving accuracy compared to uncompressed baselines.

**Strengths:**

1. The combination of reinforcement learning with supervised learning using a best trajectory memory is sensible and addresses the instability issues common in pure RL approaches. The alternating RL-SL updates within each epoch provide a reasonable balance between exploration and exploitation.

2. Unlike many abstractive compression methods that require expensive LLM calls iteratively, EFFCOMP uses lightweight BERT-based models for both reranking and sentence selection, resulting in competitive latency.

3. By operating at the sentence level and preserving original text, EFFCOMP avoids the fabrication issues demonstrated in Table 2, where Recomp-Abstractive and CompAct generate factually incorrect compressed contexts.

**Weaknesses:**

* The paper compares primarily against LongLLMLingua, Recomp, and CompAct, but misses critical recent prompt compression methods that have shown superior performance:
    - LLMLingua-2 : A BERT-based token-level classifier trained via data distillation, achieving 3x-6x faster compression than LLMLingua with better task-agnostic performance. This is directly comparable to EFFCOMP's approach and should be included.
    - TACO-RL: Another RL-based prompt compression method using REINFORCE for task-aware optimization, which is methodologically similar and should be compared.
    - 500xCompressor: A soft prompt method achieving 6x-480x compression ratios.

* FlashAttention compatibility: EFFCOMP requires attention scores for the reranker, making it potentially incompatible with FlashAttention. The paper should compare memory usage and latency against FlashAttention-enabled baselines.


* Missing evaluation on more complex multi-hop reasoning datasets like MuSiQue or StrategyQA, and single-document QA datasets like SQuAD 2.0.

**Questions:**

* Can the authors include comparisons with LLMLingua-2, TACO-RL, and 500xCompressor?
* How does EFFCOMP's memory usage and latency compare to baselines using FlashAttention? Can the method be adapted to work without requiring explicit attention scores?
* Can the authors provide an ablation study on the reward weight α (currently 0.95)? What happens at α=0.5, 0.7, 0.9, and how does this affect the Pareto frontier of compression vs. accuracy?

---

> ### Author Response · Authors · 2025-11-22
> **Response to Reviewer xN74 (1/3)**
>
> Dear Reviewer,
>
> We greatly appreciate your time and effort in reviewing our paper. Here, we would like to address your concerns point by point:
>
> > **Comment**
> > - The paper compares primarily against LongLLMLingua, Recomp, and CompAct, but misses critical recent prompt compression methods that have shown superior performance:
> > - - LLMLingua-2 : A BERT-based token-level classifier trained via data distillation, achieving 3x-6x faster compression than LLMLingua with better task-agnostic performance. This is directly comparable to EFFCOMP's approach and should be included.
> > - - TACO-RL: Another RL-based prompt compression method using REINFORCE for task-aware optimization, which is methodologically similar and should be compared.
> > - - 500xCompressor: A soft prompt method achieving 6x-480x compression ratios.
> > - Missing evaluation on more complex multi-hop reasoning datasets like MuSiQue or StrategyQA, and single-document QA datasets like SQuAD 2.0.
> > - Can the authors include comparisons with LLMLingua-2, TACO-RL, and 500xCompressor?
>
>
> Our evaluation focuses primarily on context-aware, hard prompt compression methods, as this setting is most aligned with the design goals of EFFCOMP. In general, task-agnostic compressors (e.g., LLMLingua-2) tend to yield lower QA accuracy compared to context-aware approaches, which is why our initial set of baselines emphasized methods such as LongLLMLingua, Recomp, and CompAct.
>
>
> Although we agree that including LLMLingua-2, TACO-RL, and 500xCompressor would further strengthen the empirical comparison, there are some practical obstacles:
>
>
> - TACO-RL does not release a reproducible codebase for training or evaluation, making a fair comparison infeasible within our time constraints.
>
>
> - 500xCompressor requires proprietary pretraining data or pre-trained model checkpoints that are not yet publicly available, which prevents direct benchmarking.
>
>
> Nevertheless, we conducted additional experiments with LLMLingua-2 as well as DAC [1] and CPC [2] to ensure that our approach outperforms recent baselines. Following your suggestion, we expanded our evaluation to include more diverse reasoning datasets: MuSiQue (complex multi-hop) and SQuAD 1.1 (single-document QA) in addition to HotpotQA (multi-hop) reported in the paper. Besides compression ratio, accuracy, and F1, we also reported F1_BERTScore [3] to reflect quality of the answers beyond token-level matching.
>
>
> Results of Compression Framework Evaluation Using Gemma-2-9B-IT on 30 documents
>
> | Method | HotpotQA |     |     |      | Musique |     |     |      | SQuAD |     |     |      |
> |--------|----------|-----|-----|------|---------|-----|-----|------|-------|-----|-----|------|
> |        | Comp     | Acc | F1  | BERT | Comp    | Acc | F1  | BERT | Comp  | Acc | F1  | BERT |
> | Raw documents | 1x | 34.6 | 42.3 | 73.5 | 1x | 8.8 | 14.8 | 63.8 | 1x | 40.3 | 45.3 | **74.2** |
> | CPC | 5.6x | 30.5 | 36.9 | 70.4 | 5.3x | 9.2 | 16.0 | 62.8 | 5.1x | 42.4 | 45.5 | 72.7 |
> | LongLLMLingua | 5.1x | 31.8 | 38.2 | 70.6 | 5.1x | 9.4 | 15.4 | 61.8 | 5x | 40.1 | 44.3 | 72.3 |
> | DAC | 5.1x | 24.7 | 32.3 | 68.1 | 5.1x | 6.4 | 12.9 | 60.2 | 5.3x | 25.4 | 32.4 | 67.2 |
> | LLMLingua2 | 5.4x | 25.0 | 32.6 | 68.9 | 5.4x | 7.4 | 14.0 | 63.1 | 5.3x | 26.5 | 31.7 | 67.8 |
> | Our method (No reranker) | 26.8x | **37.3** | **44.2** | **74.0** | 6.0x | 7.6 | 13.8 | 62.3 | 36.2x | **43.3** | **47.1** | 74.0 |
> | Our method (BGE) | **39.5x** | 36.4 | 43.1 | 72.6 | 9.9x | 8.7 | 15.4 | 63.1 | **42.2x** | 41.3 | 44.6 | 72.4 |
> | Our method (BGE+OOD) | - | - | - | - | **16.8x** | **10.8** | **17.6** | **64.1** | 32.1x | 40.3 | 43.5 | 72.1 |
>
> Interestingly, for MuSiQue, our method performs better under the OOD setting (trained on HotpotQA and tested on MuSiQue). This is likely because the fine-tuning set of MuSiQue is very limited (657 samples, compared to over 10,000 samples for the other datasets). Nevertheless, our method remains the top performer across all compression framework baselines.
>
> [1] Yi Zhao, et al., DAC: A Dynamic Attention-aware Approach for Task-Agnostic Prompt Compression, ACL 2025.
>
>
> [2] Liskavets, et al., Prompt Compression with Context-Aware Sentence Encoding for Fast and Improved LLM Inference, AAAI 2025.
>
>
> [3] Zhange, et al., BERTScore: Evaluating Text Generation with BERT, ICLR 2020

---

> ### Author Response · Authors · 2025-11-22
> **Response to Reviewer xN74 (2/3)**
>
> > **Comment**
> > - FlashAttention compatibility: EFFCOMP requires attention scores for the reranker, making it potentially incompatible with FlashAttention. The paper should compare memory usage and latency against FlashAttention-enabled baselines.
> > - How does EFFCOMP's memory usage and latency compare to baselines using FlashAttention? Can the method be adapted to work without requiring explicit attention scores?
>
> From our understanding, EFFCOMP does not require access to raw attention scores from the reranker. Instead, EFFCOMP only requires the reranker’s predicted relevance scores to select the top-k documents. Therefore, EFFCOMP is compatible with FlashAttention as long as the underlying model supports it.
>
>
> We also need to mention that, after revisiting our evaluation code, we found an issue that caused latency to be overestimated for all methods. We have corrected this problem in our implementation, and we will incorporate the updated results into the revised version of the paper that will be uploaded soon.
>
>
> We agree that comparing memory usage and latency against FlashAttention-enabled baselines is valuable. To address this, we enabled FlashAttention for all compression frameworks that natively support it.
>
>
> Finally, We conducted additional experiments to report both latency and memory usage under FlashAttention-enabled configurations:
>
>
> - Latency results (in seconds):
>
>
> Latency table
> | Method | Compressor Latency (s) | LLM Inference Latency (s) | Total Latency (s) |
> |--------|-------------------------|-----------------------------|--------------------|
> | Raw documents | - | 1.84 | 1.84 |
> | LongLLMLingua (Flash attention) | 2.31 | 0.90 | 3.21 |
> | Recomp Extractive | 0.19 | 0.74 | 0.93 |
> | Recomp Abstractive | 2.84 | **0.71** | 3.55 |
> | Compact (Flash attention) | 16.3 | 0.72 | 17.02 |
> | EffComp (BGE) (Flash attention at sentence selector) | 0.08 | 0.82 | 0.90 |
> | EffComp (Jina) (Flash attention full pipeline) | **0.04** | 0.82 | **0.86** |
> | EffComp (GTE) (Flash attention full pipeline) | 0.12 | 0.82 | 0.94 |
>
>
> - Memory usage results:
>
>
> Memory table
> |Method|Compressor Memory (MB)|LLM inference memory (MB)|
> |-------------------|-----------------|----------|
> |Raw documents|-|11,874.37|
> |LongLLMLingua (Flash attention)|16,129.06|6,935.18|
> |Recomp Extractive|**945.46**|6,619.45|
> |Recomp Abstractive|5,934.80|8,208.77|
> |Compact (Flash attention)|14,090.37|**6,580.15**|
> |EffComp (BGE) (Flash attention at sentence selector)|1,252.01|7,202.59|
> |EffComp (Jina) (Flash attention full pipeline)|1,242.26|7,141.47|
> |EffComp (GTE) (Flash attention full pipeline)|1,323.48|7,178.40|
>
>
> It can be observed that our method is the most efficient method in terms of compression latency and is competitive in terms of memory consumption.
>
>
> (Latency is measured by running the entire pipeline, including tokenization, model forward pass, and detokenization.)
>
>
> (Memory usage is reported as the average peak GPU memory during the model forward pass. For Recomp Abstractive, since it does not support per-sample measurement, we report the peak memory across 500 examples.)

---

> ### Author Response · Authors · 2025-11-22
> **Response to Reviewer xN74 (3/3)**
>
> > **Comment**
> > - Can the authors provide an ablation study on the reward weight α (currently 0.95)? What happens at α=0.5, 0.7, 0.9, and how does this affect the Pareto frontier of compression vs. accuracy?
>
>
> Purpose of α in our reward design
>
>
> The α parameter serves two roles:
>
>
>
> 1. Shaping the desired compression behavior during RL training
>
>
> α values in the range (0, 1) control the compression behavior: they reduce compression when all samples in a trajectory are incorrect and allow higher compression when at least some samples are correct.
>
>
> Because GRPO applies Z-score normalization to advantages, different α values in (0,1) end up with similar normalized advantage scales, meaning α does not substantially change the gradient dynamics during training.
>
>
> Furthermore, our supervised finetuning stage already biases the model heavily toward accuracy correct samples that have the highest compression ratios receive the highest rewards and dominate policy updates. As a result, the RL behavior (reducing compression when incorrect, increasing when correct) remains consistent across reasonable α values.
>
>
> 2. Selecting the best model along the compression–accuracy trade-off
>
>
>
> We conduct experiments using four values of α (0.65 , 0.75 , 0.85 , 0.95) for 3 epochs to better understand the trade-offs between accuracy, reward, and compression rate. Rewards are evaluated on 1,000 validation samples.
>
>
> |Epoch|Reward|Accuracy|Compression rate (**α = 0.65**)|
> |---|---|---|---|
> |**0**|**81.24**|**90**|**81.2**|
> |1|80.12|91.8|70.6|
> |2|80.61|91.2|74.3|
>
>
> |Epoch|Reward|Accuracy|Compression rate (**α = 0.75**)|
> |---|---|---|---|
> |**0**|**83.44**|**90.6**|**76.65**|
> |1|81.86|90.2|71.78|
> |2|82.14|90.6|71.18|
>
>
>
> |Epoch|Reward|Accuracy|Compression rate (**α = 0.85**)|
> |---|---|---|---|
> |0|85.85|89.8|78.87|
> |**1**|**86.54**|**90.5**|**79.33**|
> |2|86.47|91.4|71.77|
>
>
>
> |Epoch|Reward|Accuracy|Compression rate (**α = 0.95**)|
> |---|---|---|---|
> |0|89.26|90.6|79.40|
> |**1**|**90.81**|**92.2**|**76.81**|
> |2|89.03|90.7|70.65|
>
> In the validation phase, we observe that for α = 0.95, the checkpoint with the highest reward also consistently corresponds to the checkpoint with the highest accuracy. In contrast, for smaller α values, the highest reward is primarily driven by achieving higher compression rates. This indicates that as α increases, the reward aligns more closely with accuracy, whereas lower α values cause the reward to favor more aggressive compression.
>
>
> These results illustrate that α provides a controllable trade-off: higher α values favor accuracy, while lower α values encourage stronger compression. This offers users the flexibility to select checkpoints that best match their preferred balance between accuracy and compression within the validation set.
>
>
> And we evaluate the selected highest reward models on hotpotQA Dataset
>
>
> | Method        | Alpha | Comp  | Acc  | F1    |
> |---------------|-------|-------|------|-------|
> | EffComp (BGE) | 0.65  | **47.2x** | 35.7 | 42.6  |
> | EffComp (BGE) | 0.75  | 33.1x | 36.0 | 42.6  |
> | EffComp (BGE) | 0.85  | 46.4x | 36.2 | 42.9  |
> | EffComp (BGE) | 0.95  | 39.5x | **36.4** | **43.1** |
>
>
> Importantly, we observe that the relationship between accuracy and compression rate on the test set is consistent with the behavior seen during validation. Although our hybrid RL+SL training already prioritizes accuracy, using a higher α is still necessary to ensure that model selection aligns with our context-aware compression objective, where accuracy remains primary and compression is secondary.
>
>
> And thank you for raising this point. After revisiting our paper, we acknowledge that the description of α may be misleading, and we will revise it soon to improve clarity.

---

### Author Response · Authors · 2025-11-26
**General Response to All Reviewers**

Dear Reviewer,

Thank you for your helpful comments. We’ve updated the paper to address all of your suggestions. The revised version now includes:

Add Evaluation: LongLLMLingua at 19× Compression Ratio — Lines 332, 1150

Add Citation: Training-Free Attention-Based Methods — Lines 42, 502

Add Motivation: Why a Hybrid Method Is Needed — Line 181

Add Explanation: Why We Need α — Line 210

Add Citation: Ablation Studies on RL vs. SL — Line 294

Update RQ2: Latency with FlashAttention-2 and Pipeline (and how to measure it) — Lines 420, 432

Add Input Length to Latency Table — Line 432

Add RQ3: Do Extractive Methods Introduce Fewer Factual Inconsistencies? — Line 427

Add RQ4: Can Phase II GRPO effectively correct the biased sentence-selection behavior introduced by Phase I pretraining? — Line 462

Add Semantic-Based Evaluation Using BERTScore — Line 1180

Add Appendix: Additional Comparison with Other Compression Methods — Line 1293

Add Memory Usage with FlashAttention-2 (and how to measure it) — Lines 1309, 1336

Add Appendix: Information Efficiency Analysis — Line 1346

Add Appendix: Training Time per Epoch — Line 1453

Add Appendix : Impact of the Weighting Factor α - Line 1501

We believe these revisions significantly improve the clarity and overall quality of the paper.
Thank you again for your valuable feedback.

---

### Note · Authors · 2026-01-05

I have read and agree with the venue's withdrawal policy on behalf of myself and my co-authors.